# Efficient active learning of sparse halfspaces with arbitrary bounded noise

**Chicheng Zhang**
University of Arizona
chichengz@cs.arizona.edu

**Jie Shen**
Stevens Institute of Technology
jie.shen@stevens.edu

**Pranjal Awasthi**
Google Research and Rutgers University
pranjalawasthi@google.com

## Abstract

We study active learning of homogeneous $s$-sparse halfspaces in $\mathbb{R}^d$ under the setting where the unlabeled data distribution is isotropic log-concave and each label is flipped with probability at most $\eta$ for a parameter $\eta \in \left[0, \frac{1}{2}\right)$, known as the bounded noise. Even in the presence of mild label noise, i.e. $\eta$ is a small constant, this is a challenging problem and only recently have label complexity bounds of the form $\tilde{O}\left(s \cdot \mathrm{polylog}\left(d, \frac{1}{\epsilon}\right)\right)$ been established in [Zhang 2018] for computationally efficient algorithms. In contrast, under high levels of label noise, the label complexity bounds achieved by computationally efficient algorithms are much worse: the best known result [Awasthi et al. 2016] provides a computationally efficient algorithm with label complexity $\tilde{O}\left(\left(\frac{s \ln d}{\epsilon}\right)^{2^{\mathrm{poly}(1/(1-2\eta))}}\right)$, which is label-efficient only when the noise rate $\eta$ is a fixed constant. In this work, we substantially improve on it by designing a polynomial time algorithm for active learning of $s$-sparse halfspaces, with a label complexity of $\tilde{O}\left(\frac{s}{(1-2\eta)^4} \mathrm{polylog}\left(d, \frac{1}{\epsilon}\right)\right)$. This is the first efficient algorithm with label complexity polynomial in $\frac{1}{1-2\eta}$ in this setting, which is label-efficient even for $\eta$ arbitrarily close to $\frac{1}{2}$. Our active learning algorithm and its theoretical guarantees also immediately translate to new state-of-the-art label and sample complexity results for full-dimensional active and passive halfspace learning under arbitrary bounded noise.

## 1 Introduction

In machine learning and statistics, linear classifiers (i.e. halfspaces) are arguably one of the most important models as witnessed by a long-standing research effort dedicated to establishing computationally efficient and provable algorithms for halfspace learning [66, 81, 23]. In practical applications, however, data are often corrupted by various types of noise [74, 14], are expensive to annotate [22, 27], and are of high or even infinite dimensions [13, 17]. These characteristics rooted in contemporary machine learning problems pose new challenges to the design and analysis of learning algorithms for halfspaces. As a result, there has been extensive study of *noise-tolerant*, *label-efficient*, and *attribute-efficient* algorithms in the last few decades.

**Noise-tolerant learning.** In the noiseless setting where there is a halfspace that has zero error rate with respect to the data distribution, it is well known that by simply finding a halfspace that fits all the training examples using linear programming, one is guaranteed vanishing generalization error [82]. In the presence of data corruption, the success of efficient learning of halfspaces crucially depends on the underlying noise model. For instance, [14] proposed a polynomial time algorithm

that provably learns halfspaces when the labels are corrupted by random classification noise, that is, each label is flipped independently with a fixed probability $\eta \in \left[0, \frac{1}{2}\right)$. The bounded noise model, also known as Massart noise [74, 75, 59], is a significant generalization of the random classification noise model, in that the adversary is allowed to flip the label of each example $x$ with a different probability $\eta(x)$, with the only constraint $\eta(x) \leq \eta$ for a certain parameter $\eta \in \left[0, \frac{1}{2}\right)$. Due to its highly asymmetric nature, it remains elusive to develop computationally efficient algorithms that are robust to bounded noise. As a matter of fact, the well-known averaging scheme [47] and one-shot convex loss minimization are both unable to guarantee excess error arbitrarily close to zero even with infinite supply of training examples [3, 4, 29]. Therefore, a large body of recent works are devoted to designing more sophisticated algorithms to tolerate bounded noise, see, for example, [3, 4, 91, 86, 88, 29, 31].

**Label-efficient learning.** Motivated by many practical applications in which there are massive amounts of unlabeled data that are expensive to annotate, active learning was proposed as a paradigm to mitigate labeling costs [22, 26]. In contrast to traditional supervised learning (also known as passive learning) where the learner is presented with a set of labeled training examples, in active learning, the learner starts with a set of unlabeled examples, and is allowed to make label queries during the learning process [22, 25]. By adaptively querying examples whose labels are potentially most informative, a classifier of desired accuracy can be actively learned while requiring substantially less label feedback than that of passive learning under broad classes of data distributions [41, 10].

**Attribute-efficient learning.** With the unprecedented growth of high-dimensional data generated in biology, economics, climatology, and other fields of science and engineering, it has become ubiquitous to leverage extra properties of the data into algorithmic design for more sample-efficient learning [54]. On the computational side, the goal of attribute-efficient learning is to find a *sparse* model that identifies most useful features for prediction [34]. On the statistical side, the focus is on answering when and how learning of a sparse model will lead to improved performance guarantee on sample complexity, generalization error, or mistake bound. These problems have been investigated for a long term, and the sparsity assumption proves to be useful for achieving non-trivial guarantees [13, 77, 20]. The idea of attribute-efficient learning was also explored in a variety of other settings, including online classification [54], learning decision lists [67, 68, 52, 55], and learning parities and DNFs [35].

In this work, we consider computationally efficient learning of halfspaces in all three aspects above. Specifically, we study active learning of sparse halfspaces under the more-realistic bounded noise model, for which there are a few recent works that are immediately related to ours but under different degrees of noise tolerance and distributional assumptions. In the membership query model [1], where the learner is allowed to synthesize intances for label queries, [19] proposed an algorithm that tolerates bounded noise with near-optimal $\tilde{O}\left(\frac{d}{(1-2\eta)^2} \ln \frac{1}{\epsilon}\right)$ label complexity. In the more realistic PAC active learning model [50, 6], where the learner is only allowed to query the label of the examples drawn from the unlabeled data distribution, less progress is made towards optimal performance guarantee. Under the assumption that the unlabeled data distribution is uniform over the unit sphere, [86] proposed a Perceptron-based active learning algorithm that tolerates any noise rate of $\eta \in \left[0, \frac{1}{2}\right)$, with label complexity of $\tilde{O}\left(\frac{d}{(1-2\eta)^2} \ln \frac{1}{\epsilon}\right)$. Unfortunately, it is challenging to generalize their analysis beyond the uniform distribution, as their argument heavily relies on its symmetry. Under the broader isotropic log-concave distribution over the unlabeled data, the state-of-the-art results provide much worse label complexity bounds for the bounded noise model. Specifically, [3] showed that[1] by sequentially minimizing a series of localized hinge losses, an algorithm can tolerate bounded noise up to a constant noise rate $\eta \approx 2 \times 10^{-6}$. Furthermore, [4] combined polynomial regression [45] and margin-based sampling [7] to develop algorithms that tolerate $\eta$-bounded noise for any $\eta \in \left[0, \frac{1}{2}\right)$. However, their label complexity scales as $\tilde{O}\left(d^{2^{\text{poly}(1/(1-2\eta))}} \cdot \ln \frac{1}{\epsilon}\right)$, which is exponential in $\frac{1}{1-2\eta}$ and is polynomial in $d$ only when $\eta$ is away from $\frac{1}{2}$ by a constant. This naturally raises our *first question*: can we design a computationally efficient algorithm for active learning of halfspaces, such that for any $\eta \in \left[0, \frac{1}{2}\right)$, it has a $\text{poly}\left(d, \ln \frac{1}{\epsilon}, \frac{1}{1-2\eta}\right)$ label complexity under the more general isotropic log-concave distributions?

Table 1: A comparison of our result to prior state-of-the-art works on efficient active learning of sparse halfspaces with $\eta$-bounded noise, where the unlabeled data distribution is isotropic log-concave.

| Work | Tolerates any $\eta \in \left[0, \frac{1}{2}\right)$? | Label complexity |
|------|------|------|
| [88] | ✗ | $\tilde{O}\left(s \cdot \text{polylog}\left(d, \frac{1}{\epsilon}\right)\right)$ for small enough $\eta$ |
| [4] | ✓ | $\tilde{O}\left(\left(\frac{s \ln d}{\epsilon}\right)^{2^{\text{poly}(1/(1-2\eta))}}\right)$ |
| **This work** | ✓ | $\tilde{O}\left(\frac{s}{(1-2\eta)^4} \text{polylog}\left(d, \frac{1}{\epsilon}\right)\right)$ |

Compared to the rich literature of active learning of general non-sparse halfspaces, there are relatively few works on active learning of halfspaces that both exploit the sparsity of the target halfspace and are tolerant to bounded noise. Under the assumption that the Bayes classifier is an $s$-sparse halfspace (where $s \ll d$), a few active learning algorithms have been developed. In the membership query model, a composition of the support recovery algorithm developed in [42] with the full-dimensional active learning algorithm [19] yields a procedure that can tolerate $\eta$-bounded noise with information-theoretic optimal $\tilde{O}\left(\frac{s}{(1-2\eta)^2}(\ln d + \ln \frac{1}{\epsilon})\right)$ label complexity. Under the PAC active learning model where the unlabeled data distribution is isotropic log-concave, [4] presented an efficient algorithm that has a label complexity of $\tilde{O}\left(\left(\frac{s \ln d}{\epsilon}\right)^{2^{\text{poly}(1/(1-2\eta))}}\right)$. Under the additional assumption that $\eta$ is smaller than a numerical constant substantially bounded away from $\frac{1}{2}$, [88] gave an algorithm that has label complexity of $\tilde{O}\left(s \cdot \text{polylog}\left(d, \frac{1}{\epsilon}\right)\right)$. Neither of these two works obtained a label complexity bound that is polynomial in $\frac{1}{1-2\eta}$ (specifically, the latter work has no guarantees when $\eta$ is greater than a constant, say $1/4$). This raises our *second question*: if the Bayes classifier is an $s$-sparse halfspace, can we design an efficient halfspace learning algorithm which not only works for any bounded noise rate $\eta \in \left[0, \frac{1}{2}\right)$, but also enjoys a label complexity of $\text{poly}\left(s, \ln d, \ln \frac{1}{\epsilon}, \frac{1}{1-2\eta}\right)$?

## 1.1 Summary of our contributions

In this work, we answer both of the above questions in the affirmative. Specifically, we focus on the setting where the unlabeled data are drawn from an isotropic log-concave distribution, and the label noise satisfies the $\eta$-bounded noise condition for any $\eta \in \left[0, \frac{1}{2}\right)$. We develop an attribute-efficient learning algorithm that runs in polynomial time, and achieves a label complexity of $\tilde{O}\left(\frac{s}{(1-2\eta)^4} \cdot \text{polylog}\left(d, \frac{1}{\epsilon}\right)\right)$ provided that the underlying Bayes classifier is an $s$-sparse halfspace. Our results therefore substantially improve upon the state-of-the-art label complexity of $\tilde{O}\left(\left(\frac{s \ln d}{\epsilon}\right)^{2^{\text{poly}(1/(1-2\eta))}}\right)$ in the same setting [4]. Even in the non-sparse setting (by letting $s = d$), our label complexity bound $\tilde{O}\left(\frac{d}{(1-2\eta)^4} \cdot \ln \frac{1}{\epsilon}\right)$ is the first one of order $\text{poly}\left(d, \ln \frac{1}{\epsilon}, \frac{1}{(1-2\eta)}\right)$. Prior to this work, the best label complexity is $\tilde{O}\left(d^{2^{\text{poly}(1/(1-2\eta))}} \cdot \ln \frac{1}{\epsilon}\right)$ [4]. We summarize and compare our results in active learning to the state of the art in Tables 1 and 2, in the sparse and non-sparse setting, respectively.

As a side result of our main discoveries, our algorithm also achieves a state-of-the-art sample complexity of $\tilde{O}\left(\frac{d}{(1-2\eta)^3}\left(\frac{1}{(1-2\eta)^3} + \frac{1}{\epsilon}\right)\right)$ for passive learning of $d$-dimensional halfspaces under the same assumptions of noise and data distribution. In an independent and concurrent work [31], an efficient (passive) halfspace learning algorithm that tolerates $\eta$-bounded noise has been developed, under a broader family of unlabeled data distributions. Specializing their result to the setting when the unlabeled distribution is isotropic log-concave, their algorithm has a higher sample complexity of $O\left(\frac{d^9}{\epsilon^4(1-2\eta)^{10}}\right)$. We discuss the implications of our work on passive learning in Section 4.1, and additional related works in Appendix A.

## 1.2 An overview of our techniques

We discuss the main techniques we developed in this paper below.

Table 2: A comparison of our result to prior state-of-the-art works on active learning of non-sparse halfspaces with $\eta$-bounded noise, where the unlabeled data distribution is isotropic log-concave.

| Work | Tolerates any $\eta \in \left[0, \frac{1}{2}\right)$? | Label complexity |
|------|------|------|
| [3] | ✗ | $\tilde{O}\left(d \ln \frac{1}{\epsilon}\right)$ for small enough $\eta$ |
| [4] | ✓ | $\tilde{O}\big(d 2^{\text{poly}(1/(1-2\eta))} \cdot \ln \frac{1}{\epsilon}\big)$ |
| **This work** | ✓ | $\tilde{O}\left(\frac{d}{(1-2\eta)^4} \cdot \ln \frac{1}{\epsilon}\right)$ |

**1) Active learning via regret minimization.** We approach the active halfspace learning problem from a novel online learning perspective. Consider $v \in \mathbb{R}^d$, a vector that has angle at most $\theta$ with the underlying Bayes optimal halfspace $u$; our goal is to refine $v$ to $v'$, such that $v'$ has angle at most $\theta/2$ with $u$. To this end, we design an online linear optimization problem and apply online mirror descent with a sequence of linear loss functions $\{w \mapsto \langle g_t, w\rangle\}_{t=1}^T$ to refine the initial iterate $w_1 = v$. By standard results, e.g. [62, Theorem 6.8], we are guaranteed that after $T$ iterations,

$$\sum_{t=1}^T \langle w_t, g_t\rangle - \sum_{t=1}^T \langle u, g_t\rangle \leq \frac{D_R(u,v)}{\alpha} + \alpha \sum_{t=1}^T \|g_t\|_q^2,$$

where $R(\cdot)$ is a 1-strongly convex regularizer with respect to certain $\ell_p$-norm, $D_R$ is its induced Bregman divergence, $q$ is the conjugate exponent of $p$, and $\alpha$ is the step size. Our key idea is to construct an appropriate gradient $g_t$ (which depends on the random draw of the data), such that (a) $\langle w_t, g_t\rangle$ is small; and (b) $\mathbb{E}\left[\langle u, -g_t\rangle \mid w_t\right] \geq f_{u,b}(w_t)$ for some function $f_{u,b}(w)$ that measures the distance between the input vector $w_t$ and $u$. We then show that the average of all $f_{u,b}(w_t)$ is small, provided that the $\ell_q$-norm of $g_t$'s are upper bounded by some constant. With a (nonstandard) online-to-batch conversion [18], we obtain a classifier $v'$ from the iterates $\{w_t\}_{t=1}^T$, such that $v'$ and $u$ has angle at most $\theta/2$. We carefully choose $(p,q)$ and a sampling scheme such that both attribute efficiency and convergence are guaranteed. See Theorem 4 for a more precise statement.

**2) A new update rule that tolerates bounded noise.** As discussed above, a key step in the above regret minimization argument is to define the gradient $g_t$ such that $\mathbb{E}\left[\langle u, -g_t\rangle \mid w_t\right] \geq f_{u,b}(w_t)$. For each iterate $w_t$, we choose to sample labeled example $(x_t, y_t)$ from the data distribution $D$ conditioned on the band $\left\{x : \left|\left\langle \frac{w_t}{\|w_t\|}, x\right\rangle\right| \leq b\right\}$, similar to [27]. Based on labeled example $(x_t, y_t)$, a natural choice is $g_t = -\mathbf{1}(\hat{y}_t \neq y_t)y_t x_t$, i.e. the negative Perceptron update, where $\hat{y}_t = \text{sign}\left(\langle w_t, x_t\rangle\right)$. Unfortunately, due to the asymmetry of the unlabeled data distribution[2], it does not have the property we desire (in fact, the induced $f_{u,b}(w_t)$ can be negative with such choice of $g_t$). To cope with this challenge, we propose a novel setting of $g_t$ that takes into account the bounded noise rate $\eta$:

$$g_t = -\mathbf{1}(\hat{y}_t \neq y_t)y_t x_t - \eta \hat{y}_t x_t = \left(-\frac{1}{2}y_t + \left(\frac{1}{2} - \eta\right)\hat{y}_t\right)x_t.$$

Observe that the above choice of $g_t$ is more aggressive than the Perceptron update, in that when $\eta > 0$, even if the current prediction $\hat{y}_t$ matches the label returned by the oracle, we still update the model. In the extreme case that $\eta = 0$, we recover the Perceptron update. We show that, this new setting of $g_t$, in conjunction with the aforementioned adaptive sampling scheme, yields a function $f_{u,b}(w)$ that possesses desirable properties. We refer the reader to Lemma 6 for a precise statement.

**3) Averaging-based initialization that exploits sparsity.** The above arguments suffice to establish a local convergence guarantee, i.e. given a vector $\tilde{v}_0$ with $\theta(\tilde{v}_0, u) \leq \frac{\pi}{32}$, one can repeatedly run a sequence of online mirror descent updates and online-to-batch conversions, such that for each $k \geq 0$, we obtain a vector $\tilde{v}_k$ such that $\theta(\tilde{v}_k, u) \leq \frac{\pi}{32 \cdot 2^k}$. It remains to answer the question of how to obtain such $\tilde{v}_0$ using active learning in an attribute-efficient manner. To this end, we design an initialization procedure that finds such $\tilde{v}_0$ with $\tilde{O}\left(\frac{s}{(1-2\eta)^4} \cdot \text{polylog}(d)\right)$ labeled examples. It consists of two stages. The first stage performs the well known averaging scheme [47], in combination with a novel

hard-thresholding step [15]. This stage gives a unit vector $w^{\sharp}$ such that $\langle w^{\sharp}, u \rangle \geq \Omega(1 - 2\eta)$ with high probability, using $\tilde{O}\left(\frac{s \ln d}{(1-2\eta)^2}\right)$ labeled examples. The second stage performs online mirror descent update with adaptive sampling as before, but with the important constraint that $\langle w_t, w^{\sharp} \rangle \geq \Omega(1 - 2\eta)$ for all iterates $w_t$. Through a more careful analysis using the function $f_{u,b}$ discussed above (that accounts for the case where input $w_t$ can have a large obtuse angle with $u$), we obtain a vector $\tilde{v}_0$ that has the desired angle upper bound, with the aforementioned label complexity. We refer the reader to Lemma 14 and Theorem 3 for more details.

## 2 Preliminaries

**Active learning in the PAC model.** We consider active halfspace learning in the agnostic PAC learning model [50, 6]. In this setting, there is an instance space $\mathcal{X} = \mathbb{R}^d$ where all examples' features take value from, and a label space $\mathcal{Y} = \{-1, 1\}$ where all examples' labels take value from. The data distribution $D$ is a joint probability distribution over $\mathcal{X} \times \mathcal{Y}$. We denote by $D_X$ the marginal distribution of $D$ on $\mathcal{X}$, and by $D_{Y|X=x}$ the conditional distribution of $Y$ given $X = x$. We will also refer to $D_X$ as unlabeled data distribution. Throughout the learning process, the active learner is given access to two oracles: EX, an unlabeled example oracle that returns $x$ randomly drawn from $D_X$, and $\mathcal{O}$, a labeling oracle takes $x$ as input and returns a label $y \sim D_{Y|X=x}$.

A classifier is a mapping from $\mathcal{X}$ to $\mathcal{Y}$. We consider halfspace classifiers of the form $h_w : x \mapsto \text{sign}(w \cdot x)$ where $\text{sign}(z) = +1$ if $z \geq 0$ and equals $-1$ otherwise. The vector $w \in \mathbb{R}^d$ is the parameter of $h_w$, which has unit $\ell_2$-norm. For a given classifier $h_w$, we measure its performance by $\text{err}(h_w, D) := \mathbb{P}_{(x,y) \sim D}(h_w(x) \neq y)$, i.e. the probability that a random example gets misclassified.

We are interested in developing active halfspace learning algorithms that achieve the agnostic PAC guarantee. Specifically, we would like to design an algorithm $\mathcal{A}$, such that it receives as inputs excess error parameter $\epsilon \in (0, 1)$ and failure probability parameter $\delta \in (0, 1)$, and with probability $1 - \delta$, after making a number of queries to EX and $\mathcal{O}$, $\mathcal{A}$ returns a halfspace $h_w$ such that $\text{err}(h_w, D) - \min_{w'} \text{err}(h_{w'}, D) \leq \epsilon$. In addition, we would like our active learner to make as few label queries as possible. We denote by $n_{\mathcal{A}}(\epsilon, \delta)$ the number of label queries of $\mathcal{A}$ given parameters $\epsilon$ and $\delta$; this is also called the *label complexity* of $\mathcal{A}$.

We will focus on sampling unlabeled examples from $D_X$ conditioned on a subset $B$ of $\mathbb{R}^d$; this can be done by rejection sampling, where we repeatedly call EX until we see an unlabeled example $x$ falling in $B$. Given a unit vector $\hat{w}$ and $b > 0$, define $B_{\hat{w},b} = \{x \in \mathbb{R}^d : |\hat{w} \cdot x| \leq b\}$. Denote by $D_{X|\hat{w},b}$ (resp. $D_{\hat{w},b}$) the distribution of $D_X$ (resp. $D$) conditioned on the event that $x \in B_{\hat{w},b}$.

**Vectors.** Let $w$ be a vector in $\mathbb{R}^d$. The $\ell_0$-"norm" $\|w\|_0$ counts its number of nonzero elements, and $w \in \mathbb{R}^d$ is said to be $s$-sparse if $\|w\|_0 \leq s$. Given $s \in \{1, \ldots, d\}$, the hard thresholding operation $\mathcal{H}_s(w)$ zeroes out all but $s$ largest (in magnitude) entries of $w$. For a scalar $\gamma \geq 1$, denote by $\|w\|_\gamma$ the $\ell_\gamma$-norm of the vector $w$. If not explicitly mentioned, $\|\cdot\|$ denotes the $\ell_2$-norm. We denote by $\hat{w} = \frac{w}{\|w\|}$ as the $\ell_2$-normalization of $w \in \mathbb{R}^d$. For two vectors $w_1, w_2$, we write $\theta(w_1, w_2) = \arccos(\hat{w}_1 \cdot \hat{w}_2)$ as the angle between them.

**Convexity.** Given a convex and differentiable function $f$, its induced Bregman divergence is given by $D_f(w, w') \overset{\text{def}}{=} f(w) - f(w') - \langle \nabla f(w'), w - w' \rangle$. Note that by the convexity of $f$, $D_f(w, w') \geq 0$ for all $w$ and $w'$. A function $f$ is said to be $\lambda$-strongly convex with respect to the norm $\|\cdot\|_\gamma$, if $D_f(w, w') \geq \frac{\lambda}{2}\|w - w'\|_\gamma^2$ holds for all $w$ and $w'$ in its domain. In our algorithm, we will use the following convex function: $\Phi_v(w) \overset{\text{def}}{=} \frac{1}{2(p-1)}\|w - v\|_p^2$, where $v$ is a reference vector in $\mathbb{R}^d$. Throughout the paper, we reserve $p$ for a specific value $p = \frac{\ln(8d)}{\ln(8d)-1}$, and reserve $q = \ln(8d)$ (note that $p^{-1} + q^{-1} = 1$). As $p \in (1, 2]$, $\Phi_v$ is 1-strongly convex with respect to $\|\cdot\|_p$ [69, Lemma 17]. In addition, $\nabla \Phi_v$ is a one-to-one mapping from $\mathbb{R}^d$ to $\mathbb{R}^d$, and hence has an inverse, denoted as $\nabla \Phi_v^{-1}$.

**Distributional assumptions.** Without distributional assumptions, it is known that agnostically learning halfspaces is computationally hard [36, 40]. We make the following two assumptions.

**Assumption 1.** *The data distribution $D$ satisfies the $\eta$-bounded noise condition with respect to an $s$-sparse unit vector $u \in \mathbb{R}^d$, where the noise rate $\eta \in [0, 1/2)$. Namely, for all $x \in \mathcal{X}$, $\mathbb{P}(y \neq \text{sign}(u \cdot x) | X = x) \leq \eta$.*

**Assumption 2.** *The unlabeled data distribution $D_X$ is isotropic log-concave over $\mathbb{R}^d$, i.e. $D_X$ has a probability density function $f$ over $\mathbb{R}^d$ such that $\ln f(x)$ is concave, and $\mathbb{E}_{x \sim D_X}\left[ xx^\top \right] = I_{d \times d}$.*

Assumption 1 implies that the Bayes optimal classifier with respect to the distribution $D$ is $h_u$. As a consequence, the optimal halfspace is $h_u$, namely $\mathrm{err}(h_u, D) = \min_{w'} \mathrm{err}(h_w, D)$. Assumption 2 has appeared in many prior works [51, 9, 5, 88]. [9] showed the following important lemma.

**Lemma 1.** *Suppose that Assumption 2 is satisfied. There exist absolute constants $c_1$ and $c_2$, such that for any two vectors $v_1$ and $v_2$,*

$$c_1 \mathbb{P}_{x \sim D_X}\left( \mathrm{sign}\left( v_1 \cdot x \right) \neq \mathrm{sign}\left( v_2 \cdot x \right) \right) \leq \theta(v_1, v_2) \leq c_2 \mathbb{P}_{x \sim D_X}\left( \mathrm{sign}\left( v_1 \cdot x \right) \neq \mathrm{sign}\left( v_2 \cdot x \right) \right).$$

## 3 Main algorithm

We present Algorithm 1, our noise-tolerant attribute-efficient active learning algorithm, in this section. It consists of two stages: an initialization stage INITIALIZE (line 2) and an iterative refinement stage (lines 3 to 5). In the initialization stage, we aim to find a vector $\tilde{v}_0$ such that $\theta(\tilde{v}_0, u) \leq \frac{\pi}{32}$; in the iterative refinement stage, we aim to bring our iterate $\tilde{v}_k$ closer to $u$ after each phase $k$. Specifically, suppose that $\theta(\tilde{v}_{k-1}, u) \leq \frac{\pi}{32 \cdot 2^{k-1}}$ at the beginning of iteration $k$, then after the execution of line 5, we aim to obtain a new iterate $\tilde{v}_k$ such that $\theta(\tilde{v}_k, u) \leq \frac{\pi}{32 \cdot 2^k}$ with high probability. The iterative refinement stage ends when $k$ reaches $k_0$, in which case we are guaranteed that $\tilde{u} = \tilde{v}_{k_0}$ is such that $\theta(\tilde{u}, u) \leq \frac{\pi}{32 \cdot 2^{k_0}} \leq c_1 \epsilon$, where $c_1$ is the constant defined in Lemma 1. From Lemma 1, we have that $\mathbb{P}_{x \sim D_X}(h_{\tilde{u}}(x) \neq h_u(x)) \leq \epsilon$. Consequently, by triangle inequality, we have that $\mathrm{err}(h_{\tilde{u}}, D) - \mathrm{err}(h_u, D) \leq \mathbb{P}_{x \sim D_X}(h_{\tilde{u}}(x) \neq h_u(x)) \leq \epsilon$.

---

**Algorithm 1** Main algorithm

**Input:** Target error $\epsilon$, failure probability $\delta$, bounded noise level $\eta$, sparsity $s$.
**Output:** Halfspace $\tilde{u}$ in $\mathbb{R}^d$ such that $\mathrm{err}(h_{\tilde{u}}, D) - \mathrm{err}(h_u, D) \leq \epsilon$.
 1: Let $k_0 = \lceil \log \frac{1}{c_1 \epsilon} \rceil$ be the total number of iterations, where $c_1$ is defined in Lemma 1.
 2: Let $\tilde{v}_0 \leftarrow$ INITIALIZE$(\frac{\delta}{2}, \eta, s)$. // See Algorithm 3.
 3: **for** phases $k = 1, 2, \ldots, k_0$ **do**
 4:     $v_{k-1} \leftarrow \mathcal{H}_s(\tilde{v}_{k-1})$.
 5:     $\tilde{v}_k \leftarrow$ REFINE$(v_{k-1}, \frac{\delta}{2k(k+1)}, \eta, s, \alpha_k, b_k, \mathcal{K}_k, R_k, T_k)$, where the step size $\alpha_k = \tilde{\Theta}\left( (1 - 2\eta)2^{-k} \right)$, bandwidth $b_k = \Theta\left( (1 - 2\eta)2^{-k} \right)$, constraint set

$$\mathcal{K}_k = \left\{ w \in \mathbb{R}^d : \|w - v_{k-1}\|_2 \leq \pi \cdot 2^{-k-3}, \|w\|_2 \leq 1 \right\},$$

   regularizer $R_k(w) = \Phi_{v_{k-1}}(w)$, number of iterations $T_k = O\left( \frac{s}{(1-2\eta)^2} \left( \ln \frac{d \cdot k^2 2^k}{\delta(1-2\eta)} \right)^3 \right)$.
 6: **return** $\tilde{u} \leftarrow \tilde{v}_{k_0}$.

---

**Algorithm 2** REFINE

**Input:** Initial halfspace $w_1$, failure probability $\delta'$, bounded noise level $\eta$, sparsity $s$, learning rate $\alpha$, bandwidth $b$, convex constraint set $\mathcal{K}$, regularization function $R(w)$, number of iterations $T$.
**Output:** Refined halfspace $\tilde{w}$.
 1: **for** $t = 1, 2, \ldots, T$ **do**
 2:     Sample $x_t$ from $D_{X|\hat{w}_t, b}$, the conditional distribution of $D_X$ on $B_{\hat{w}_t, b}$ and query $\mathcal{O}$ for its label $y_t$ (recall that $\hat{w}_t$ is the $\ell_2$-normalization of $w_t$).
 3:     Update: $w_{t+1} \leftarrow \arg\min_{w \in \mathcal{K}} D_R\left( w, \nabla R^{-1}\left( \nabla R(w_t) - \alpha g_t \right) \right)$, where the gradient $g_t = \left( -\frac{1}{2} y_t + \left( \frac{1}{2} - \eta \right) \hat{y}_t \right) x_t$, and $\hat{y}_t = \mathrm{sign}\left( w_t \cdot x_t \right)$.
 4: $\bar{w} \leftarrow \frac{1}{T} \sum_{t=1}^{T} \hat{w}_t$.
 5: **return** $\tilde{w} \leftarrow \frac{\bar{w}}{\|\bar{w}\|}$.

---

**The refinement procedure.** We first describe our refinement procedure, namely Algorithm 2, in detail. When used by Algorithm 1, it requires that the input $w_1$ has angle $\theta \in [0, \frac{\pi}{32}]$ with $u$, and aims to find a new $\tilde{w}$ such that it has angle $\theta/2$ with $u$. It performs iterative update on $w_t$'s (lines 1 to 3) in

---
**Algorithm 3** INITIALIZE
---
**Input:** Failure probability $\delta'$, bounded noise parameter $\eta$, sparsity parameter $s$.
**Output:** Halfspace $\tilde{v}_0$ such that $\theta(\tilde{v}_0, u) \leq \frac{\pi}{32}$.

1: $(x_1, y_1), \ldots, (x_m, y_m) \leftarrow$ draw $m$ examples iid from $D_X$, and query $\mathcal{O}$ for their labels, where $m = 81 \cdot 2^{51} \cdot \frac{s \ln \frac{8d}{\delta'}}{(1-2\eta)^2}$.

2: Compute $w_{\text{avg}} = \frac{1}{m} \sum_{i=1}^{m} x_i y_i$.

3: Let $w^\sharp = \frac{\mathcal{H}_{\tilde{s}}(w_{\text{avg}})}{\|\mathcal{H}_{\tilde{s}}(w_{\text{avg}})\|}$, where $\tilde{s} = \frac{81 \cdot 2^{38}}{(1-2\eta)^2} s$.

4: Find a point $w_1$ in the set $\mathcal{K} = \left\{ w : \|w\|_2 \leq 1, \|w\|_1 \leq \sqrt{s}, \langle w, w^\sharp \rangle \geq \frac{(1-2\eta)}{9 \cdot 2^{19}} \right\}$.

5: **return** $\tilde{v}_0 \leftarrow$ REFINE$(w_1, \frac{\delta'}{2}, \eta, s, \alpha, b, \mathcal{K}, R, T)$, where step size $\alpha = \tilde{\Theta}\left((1-2\eta)^2\right)$, bandwidth $b = \Theta\left((1-2\eta)^2\right)$, constraint set $\mathcal{K}$, regularizer $R(w) = \Phi_{w_1}(w)$, and number of iterations $T = O\left(\frac{s}{(1-2\eta)^4}\left(\ln \frac{d}{\delta'(1-2\eta)}\right)^3\right)$.

---

the following manner. Given the current iterate $w_t$, it defines a (time-varying) sampling region $B_{\hat{w}_t, b}$, samples an example $x_t$ from $D_X$ conditioned on $B_{\hat{w}_t, b}$, and queries its label $y_t$. This time-varying sampling strategy has appeared in many prior works on active learning of halfspaces, such as [27, 86].

Then, given the example $(x_t, y_t)$, it performs an online mirror descent update (line 3) with regularizer $R(w)$, along with a carefully designed update vector $-\alpha g_t$. The gradient vector

$$g_t = \left(-\frac{1}{2} y_t + \left(\frac{1}{2} - \eta\right) \hat{y}_t\right) x_t = \begin{cases} -\eta y_t x_t, & y_t = \hat{y}_t, \\ -(1-\eta) y_t x_t, & y_t \neq \hat{y}_t, \end{cases}$$

is a carefully-scaled version of $-y_t x_t$. Observe that if $\eta = 0$, i.e. the noise-free setting, our algorithm sets $g_t = -\mathbf{1}(\hat{y}_t \neq y_t) y_t x_t$, which is the gradient widely used in online classification algorithms, such as Perceptron [66], Winnow [54] and $p$-norm algorithms [39, 38]. As we shall see, this modified update is vital to the bounded noise tolerance property (Lemma 6). Observe that Algorithm 2 is computationally efficient, as each step of online mirror descent update only requires solving a convex optimization problem; specifically, $\mathcal{K}$ is a convex set, and $D_R(\cdot, \cdot)$ is convex in its first argument.

In the calls of Algorithm 2 in Algorithm 1, the constraint set $\mathcal{K}_k$ is different from the one in [88], where an additional $\ell_1$-norm constraint is used and is crucial for near-optimal dependence on the sparsity and dimension. Here the $\ell_1$-norm constraint is not vital. In fact, when invoking Algorithm 2, we use regularizer $R(w)$ of form $\Phi_v(w) = \frac{1}{2(p-1)} \|w - v\|_p^2$ for $p = \frac{\ln(8d)}{\ln(8d)-1}$, which is well known to induce attribute efficiency [39, 38]. See Appendix D for a formal treatment.

After obtaining the iterates $\{w_t\}_{t=1}^{T}$, we tailor online-to-batch conversion [18] to our problem: we take an average over the $\ell_2$-normalized $w_t$'s, and further normalize it to obtain our refined estimate $\tilde{w}$.

**The initialization procedure.** Our initialization procedure, Algorithm 3, aims to produce a vector $\tilde{v}_0$ such that $\theta(\tilde{v}_0, u) \leq \frac{\pi}{32}$. It consists of two stages. At its first stage, it generates a very coarse estimate of $u$, namely $w^\sharp$, as follows: first, we take the average of $x_i y_i$'s to obtain $w_{\text{avg}}$ (line 2); next, it performs hard-thresholding and normalization on $w_{\text{avg}}$ (line 3), with parameter $\tilde{s} = O\left(\frac{s}{(1-2\eta)^2}\right)$. As we will see, with $m = O(\tilde{s} \ln d)$ label queries, $w^\sharp$, the output unit vector of the first stage, is such that $\langle w^\sharp, u \rangle \geq \Omega(1 - 2\eta)$. At its second stage, it uses REFINE (Algorithm 2) to obtain a better estimate, with a constraint set $\mathcal{K}$ that incorporates the knowledge obtained at the first stage: for all $w$ in $\mathcal{K}$, $w$ satisfies $\langle w, w^\sharp \rangle \geq \Omega(1 - 2\eta)$. Note that $u \in \mathcal{K}$. Technically speaking, this additional linear constraint ensures that for all $w$ in $\mathcal{K}$, $\theta(w, u) \leq \pi - \Omega(1 - 2\eta)$, which gets around technical challenges when dealing with iterates $w_t$ that are nearly opposite to $u$. See Lemma 20 in Appendix E for more details.

We remark that it may be possible to prove a refined bound on $\theta(w_{\text{avg}}, u)$ smaller than, say, $\frac{\pi}{4}$, as existing lower bounds on $\theta(w_{\text{avg}}, u)$, e.g. Theorem 2 of [3], do not rule out such possibility. This could lead to a more sample-efficient initialization procedure that avoids using the above REFINE procedure with the specialized setting of constraint set $\mathcal{K}$. If this were the case, combining this with the guarantees of REFINE (Theorem 4 below) would imply an active learning algorithm with

information-theoretic near-optimal label complexity of $\tilde{O}(\frac{s}{(1-2\eta)^2} \operatorname{polylog}(d, \frac{1}{\epsilon}))$ in this setting. We leave this as an interesting open problem.

## 4 Performance guarantees

We now provide formal performance guarantees of Algorithm 1, showing that: 1) it is able to achieve any target excess error rate $\epsilon \in (0, 1)$; 2) it tolerates any bounded noise rate $\eta \in [0, 1/2)$; and 3) its label complexity has near-optimal dependence on the sparsity and data dimension, and has substantially improved dependence on the noise rate.

**Theorem 2** (Main result). *Suppose Algorithm 1 is run under a distribution $D$ such that Assumptions 1 and 2 are satisfied. Then with probability $1 - \delta$, it returns a halfspace $\tilde{u}$ such that $\operatorname{err}(h_{\tilde{u}}, D) - \operatorname{err}(h_u, D) \leq \epsilon$. Moreover, our algorithm tolerates any noise rate $\eta \in [0, 1/2)$, and asks for a total of $\tilde{O}\big(\frac{s}{(1-2\eta)^4} \operatorname{polylog}\big(d, \frac{1}{\epsilon}, \frac{1}{\delta}\big)\big)$ labels.*

The proof of this theorem consists of two parts: first, we show that with high probability, our initialization procedure returns a vector $\tilde{v}_0$ that is close to $u$, in the sense that $\|\tilde{v}_0\| = 1$ and $\theta(\tilde{v}_0, u) \leq \frac{\pi}{32}$ (Theorem 3); Second, we show that given such $\tilde{v}_0$, with high probability, our refinement procedure (lines 3 to 5) will finally return a vector $\tilde{v}_{k_0}$ that has the target error rate $\epsilon$ (Theorem 4). We defer the full proof of Theorem 2 to Appendix B. In Appendix 4.1, we discuss an extension of the theorem that establishes an upper bound on the number of unlabeled examples it encounters, and discuss its implication to supervised learning.

**Initialization step.** We first characterize the guarantees of INITIALIZE in the following theorem.

**Theorem 3** (Initialization). *Suppose Algorithm 3 is run under a distribution $D$ such that Assumptions 1 and 2 are satisfied, with noise rate $\eta \in [0, 1/2)$, sparsity parameter $s$, and failure probability $\delta'$. Then with probability $1 - \delta'$, it returns a unit vector $\tilde{v}_0$, such that $\theta(\tilde{v}_0, u) \leq \frac{\pi}{32}$. In addition, the total number of label queries it makes is $O\big(\frac{s}{(1-2\eta)^4}\big(\ln \frac{d}{\delta'(1-2\eta)}\big)^3\big)$.*

We prove the theorem in two steps: first, we show that lines 2 and 3 of Algorithm 3 returns a unit vector $w^\sharp$ that has a positive inner product with $u$, specifically, $\Omega(1 - 2\eta)$. This gives a halfspace constraint on $u$, formally $\langle w^\sharp, u \rangle \geq \Omega(1 - 2\eta)$. Next, we show that applying Algorithm 2 with the feasible set $\mathcal{K}$ that incorporates the halfspace constraint, and an appropriate choice of $b$, gives a unit vector $\tilde{v}_0$ such that $\theta(\tilde{v}_0, u) \leq \frac{\pi}{32}$. We defer the full proof of the theorem to Appendix E.

**Refinement step.** Theorem 4 below shows that after hard thresholding and one step of REFINE (line 5), Algorithm 1 halves the angle upper bound between the current predictor $\tilde{v}_k$ and $u$. Therefore, by induction, repeatedly applying Algorithm 2 ensures $\theta(\tilde{v}_{k_0}, u) = O(\epsilon)$ with high probability.

**Theorem 4** (Refinement). *Suppose we are given a unit vector $\tilde{v}$ such that $\theta(\tilde{v}, u) \leq \theta \in [0, \frac{\pi}{32}]$. Define $v \stackrel{def}{=} \mathcal{H}_s(\tilde{v})$. Suppose Algorithm 2 is run with initial halfspace $v$, confidence $\delta'$, bounded noise rate $\eta$, sparsity $s$, bandwidth $b = \Theta((1 - 2\eta)\theta)$, step size $\alpha = \tilde{\Theta}((1 - 2\eta)\theta)$, constraint set $\mathcal{K} = \{w : \|w - v\|_2 \leq 2\theta, \|w\|_2 \leq 1\}$, regularization function $R(w) = \Phi_v$, number of iterations $T = O\big(\frac{s}{(1-2\eta)^2}\big(\ln \frac{d}{\delta'\theta(1-2\eta)}\big)^3\big)$. Then with probability $1 - \delta'$, it outputs $\tilde{v}'$ such that $\theta(\tilde{v}', u) \leq \frac{\theta}{2}$; moreover, the total number of label queries it makes is $O\big(\frac{s}{(1-2\eta)^2}\big(\ln \frac{d}{\delta'\theta(1-2\eta)}\big)^3\big)$.*

The intuition behind the theorem is as follows: we define a function $f_{u,b}(w)$ that measures the closeness between unit vector $w$ and the underlying optimal classifier $u$. As Algorithm 2 performs online mirror descent on the linear losses $\{w \mapsto \langle g_t, w \rangle\}_{t=1}^T$, it achieves a regret guarantee, which implies an upper bound on the average value of $\{f_{u,b}(w_t)\}_{t=1}^T$. As $f_{u,b}(w)$ measures the closeness between $w$ and $u$, we can conclude that there is a overwhelming portion of $\{w_t\}_{t=1}^T$ that has a small angle with $u$. Consequently, by averaging and normalization, it can be argued that the resulting unit vector $\tilde{v}' = \tilde{w}$ is such that $\theta(\tilde{v}', u) \leq \frac{\theta}{2}$. We defer the full proof of the theorem to Appendix D.

### 4.1 Implication for supervised learning

In this section, we briefly outline the implication of our results to supervised learning (i.e. passive learning). As our algorithms acquire examples in a streaming fashion, it can be readily seen that,

a variant of Algorithm 1 can be viewed as a supervised learning algorithm: each time Algorithm 1 draws unlabeled example from $D_X$, the variant immediately queries $\mathcal{O}$ for its label. Consequently, the number of examples it encounters equals the total number of labeled examples it consumes, which corresponds to its sample complexity.

We now show that Algorithm 1 uses at most $\tilde{O}\left(\frac{s}{(1-2\eta)^3}\left(\frac{1}{(1-2\eta)^3}+\frac{1}{\epsilon}\right)\cdot \text{polylog}\,(d)\right)$ unlabeled examples; therefore, its induced supervised learning algorithm has a sample complexity of $\tilde{O}\left(\frac{s}{(1-2\eta)^3}\left(\frac{1}{(1-2\eta)^3}+\frac{1}{\epsilon}\right)\cdot \text{polylog}\,(d)\right)$. Without the sparsity assumption (i.e. setting $s=d$), this yields a sample complexity of $\tilde{O}\left(\frac{d}{(1-2\eta)^3}\left(\frac{1}{(1-2\eta)^3}+\frac{1}{\epsilon}\right)\right)$.

**Theorem 5.** *Suppose that Assumptions 1 and 2 are satisfied. With probability* $1-\delta$, *Algorithm 1 makes at most* $\tilde{O}\left(\frac{s}{(1-2\eta)^3}\left(\frac{1}{(1-2\eta)^3}+\frac{1}{\epsilon}\right)\cdot \text{polylog}\,(d)\right)$ *queries to the unlabeled example generation oracle* EX.

The proof of Theorem 5 can be found in Appendix C.

## 5  Conclusion and discussion

In this work we substantially improve on the state-of-the-art results on efficient active learning of sparse halfspaces under bounded noise. Furthermore, our proposed framework of online mirror descent with the margin-based analysis could have other applications in the design of learning algorithms. Our algorithm has a near-optimal label complexity of $\tilde{O}\left(\frac{s}{(1-2\eta)^2}\,\text{polylog}\,\left(d,\frac{1}{\epsilon}\right)\right)$ in the local convergence phase, while having a suboptimal label complexity of $\tilde{O}\left(\frac{s}{(1-2\eta)^4}\,\text{polylog}\,(d)\right)$ in the initialization phase. It is still an open question whether we can obtain an efficient algorithm that achieves the information-theoretically optimal label complexity of $\tilde{O}\left(\frac{s}{(1-2\eta)^2}\,\text{polylog}\,\left(d,\frac{1}{\epsilon}\right)\right)$, possibly via suitable modifications of our initialization procedure. It would be promising to extend our results beyond isotropic log-concave distributions [11], and would be interesting to investigate whether our algorithmic insights can find applications for learning halfspaces under the Tsybakov noise model [79] and the malicious noise model [80, 48].

## Broader Impact

This paper investigates a fundamental problem in machine learning and statistics. The theory and algorithms presented in this paper are expected to benefit many broad fields in science and engineering, such as learning theory, robust statistics, optimization, and applications in biology, climatology, and seismology, to name a few. Our research belongs to the general paradigm of interactive learning, in which the learning agent need to design adaptive sampling schemes to maximize data efficiency. We are well aware that one needs to be careful in designing such sampling schemes, to avoid unintended harms such as discrimination.

## Acknowledgments and Disclosure of Funding

Jie Shen is supported by NSF-IIS-1948133 and the startup funding of Stevens Institute of Technology. Chicheng Zhang would like to thank Ning Hao and Helen Hao Zhang for helpful discussions on marginal screening for variable selection, which inspired the averaging-based initialization procedure in this paper.

## Footnotes

[1][3] phrased all the results with respect to the uniform distribution of the unlabeled data. However, their analysis can be straightforwardly extended to isotropic log-concave distributions, and was spelled out in [88].

[2] [86] extensively utilizes the symmetry of the uniform unlabeled distribution to guarantee that the expectation is positive if the angle between $w_t$ and $u$ is large; we cannot use this as we are dealing with a more general family of log-concave unlabeled distribution, which can be highly asymmetric.

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
