[Supplementary Material]

# A Additional related works

There is a rich literature on learning halfspaces in the presence of noise. For instance, [14, 33] studied noise-tolerant learning of halfspaces under the random classification noise model, where each label is flipped independently with probability exactly $\eta$. Their algorithm proceeds as optimizing a sequence of modified Perceptron updates, and the analysis implies that the desired halfspace can be learned in polynomial time with respect to arbitrary unlabeled distribution. [49] considered learning halfspaces with malicious noise, where with some probability the learner is given an adversarially-generated pair of feature vector and label. Notably, their work showed that under such noise model, it is still possible to learn a good halfspace for arbitrary data distribution in polynomial time, provided that the noise rate is $\tilde{\Omega}(\frac{\epsilon}{d})$. In a series of recent work, this bound has been significantly improved by making additional assumptions on the data distribution and more sophisticated algorithmic designs [51, 56, 5]. The bounded noise, also known as Massart noise [59], was initially studied in [74, 75, 65]. Very recently, [29] presented an efficient learning algorithm that has distribution-free guarantee (albeit with vanishing excess error guarantees only in the random classification noise setting), whereas most of the prior works are built upon distributional assumptions [3, 4, 91, 86, 88]. It is worth noting that other types of noise, such as malicious noise [80] and adversarial noise [50], have also been widely studied [45, 51, 46, 24, 5, 30, 73].

There is a large body of theoretical works on active learning for general hypothesis classes; see e.g. [25, 6, 41] and the references therein. Despite their generality, many of the algorithms developed are not guaranteed to be computationally efficient. For efficient noise-tolerant active halfspace learning, aside from the aforementioned works in the main text, we also remark that the work of [8] provides the first computationally efficient algorithm for halfspace learning under log-concave distribution that tolerates random classification noise, with a label complexity of $\text{poly}\left(d, \ln\frac{1}{\epsilon}, \frac{1}{1-2\eta}\right)$. Prior to our work, it is not known how to obtain an attribute-efficient active learning algorithm with label complexity $\text{poly}\left(s, \ln d, \ln\frac{1}{\epsilon}, \frac{1}{1-2\eta}\right)$, even under this weaker random classification noise setting.

Parallel to the development of attribute-efficient learning in learning theory, there have been a large body of theoretical works developed in compressed sensing [32]. In this context, the goal is twofold: 1) design an efficient data acquisition scheme to significantly compress a high-dimensional but effectively sparse signal; and 2) implement an estimation algorithm that is capable of reconstructing the underlying signal from the measurements. These two phases are bind together in view of the need of low sample complexity (i.e. number of measurements), and a large volume of theoretical results have been established to meet the goal. For instance, many of the early works utilize linear measurements for the sake of its computational efficiency, and focus on the development of effective recovery procedures [20, 77, 17, 84, 78, 15, 60, 37, 90, 71, 72]. In its 1-bit variant [16], the linear measurements are further quantized to a binary code, and it bears the potential of savings of physical storage as long as accurate estimation in the low-bit setting does not require significantly more measurements. In order to account for the new data acquisition scheme, a large body of new estimation paradigms are developed in recent years. For instance, [44] showed that exact recovery can be achieved by seeking a global optimum of a sparsity-constrained nonconvex program. [63, 64, 89] demonstrated that $\ell_1$-norm based convex programs inherently behave as well as the nonconvex counterpart in terms of estimation error. Generally speaking, the difference between 1-bit compressed sensing and learning of halfspaces lies in the fact that in compress sensing one is able to control how the data are collected. Interestingly, [53, 12] showed that if we manually inject Gaussian noise before quantization and pass the variance parameter to the recovery algorithm, it is possible to estimate the magnitude of the signal.

The idea of active learning is also broadly explored in the compressed sensing community under the name of adaptive sensing [43, 58]. Though [2] showed that adaptive sensing strategy does not lead to significant improvement on sample complexity, a lot of recent works illustrated that it does when there are additional constraints on the sensing matrix [28], or when 1-bit quantization is applied during data acquisition [12]. As a matter of fact, [12] showed that by adaptively generating the 1-bit measurements, it is possible to design an efficient recovery algorithm that has exponential decay in reconstruction error which essentially translates into $O\left(s\log(d)\log(1/\epsilon)\right)$ sample complexity.

Noisy models are also studied in compressed sensing. For instance, [61, 21, 76] considered the situation where a fraction of the data are corrupted by outliers. [64] studied robustness of convex programs when the 1-bit measurements are either corrupted by random noise or adversarial noise.

# B  Proof of Theorem 2

In this section we present a detailed proof of Theorem 2, our main result.

*Proof of Theorem 2.* We define event $E_0$ as the event that the guarantees of Theorem 3 holds with failure probability $\delta' = \frac{\delta}{2}$. In addition, we define event $E_k$ as the event that the guarantees of Theorem 4 holds for input $\tilde{v} = \tilde{v}_{k-1}$, angle upper bound $\theta = \frac{\pi}{32 \cdot 2^{k-1}}$ and output $\tilde{v}' = \tilde{v}_k$ with failure probability $\delta' = \frac{\delta}{2k(k+1)}$. It can be easily seen that $\mathbb{P}(E_0) \geq 1 - \frac{\delta}{2}$, and $\mathbb{P}(E_k) \geq 1 - \frac{\delta}{2k(k+1)}$ for all $k \geq 1$.

Consider event $E = \bigcap_{k=0}^{k_0} E_k$. Using union bound, we have that $\mathbb{P}(E) \geq 1 - \frac{\delta}{2} - \sum_{k=1}^{k_0} \frac{\delta}{2k(k+1)} \geq 1 - \delta$. On event $E$, we now show inductively that $\theta(\tilde{v}_k, u) \leq \frac{\pi}{32 \cdot 2^k}$ for all $k \in \{0, 1, \ldots, k_0\}$.

**Base case.**  By the definition of $E_0$ and the fact that $E \subset E_0$, we have $\theta(\tilde{v}_0, u) \leq \frac{\pi}{32}$.

**Inductive case.**  Now suppose that on event $E$, we have $\theta(\tilde{v}_{k-1}, u) \leq \frac{\pi}{32 \cdot 2^{k-1}}$. Now by the definition of event $E_k$, we have that after Algorithm 2, we obtain a unit vector $v_k$ such that $\theta(\tilde{v}_k, u) \leq \frac{\pi}{32 \cdot 2^k}$.

This completes the induction. Specifically, on event $E$, after the last phase $k_0 = \lceil \log \frac{1}{c_1 \epsilon} \rceil$, we obtain a vector $\tilde{u} = \tilde{v}_{k_0}$, such that $\theta(\tilde{u}, u) \leq \frac{\pi}{32 \cdot 2^{k_0}} \leq c_1 \epsilon$. Now applying Lemma 1, we have that $\mathbb{P}(\mathrm{sign}\,(\tilde{u} \cdot x) \neq \mathrm{sign}\,(u \cdot x)) \leq \frac{1}{c_1} \theta(\tilde{u}, u) \leq \epsilon$. By triangle inequality, we conclude that

$$\mathrm{err}(h_{\tilde{u}}, D) - \mathrm{err}(h_u, D) \leq \mathbb{P}(\mathrm{sign}\,(\tilde{u} \cdot x) \neq \mathrm{sign}\,(u \cdot x)) \leq \epsilon.$$

We now upper bound the label complexity of Algorithm 1. The initialization phase uses $n_0 = O\left( \frac{s}{(1-2\eta)^4} \left( \ln \frac{d}{\delta(1-2\eta)} \right)^3 \right)$ labeled queries. Meanwhile, for every $k \in [k_0]$, Algorithm 2 at phase $k$ uses $n_k = O\left( \frac{s}{(1-2\eta)^2} \left( \ln \frac{d \cdot k^2 2^k}{\delta(1-2\eta)} \right)^3 \right)$ label queries. Therefore, the total number of label queries by Algorithm 1 is:

$$
\begin{aligned}
n = n_0 + \sum_{k=1}^{k_0} n_k &= O\left( \frac{s}{(1-2\eta)^2} \left( \frac{1}{(1-2\eta)^2} \ln \frac{d}{\delta(1-2\eta)} \right)^3 + \ln \frac{1}{\epsilon} \cdot \left( \ln \frac{d}{\delta \epsilon (1-2\eta)} \right)^3 \right) \\
&= O\left( \frac{s}{(1-2\eta)^4} \left( \ln \frac{d}{\delta \epsilon (1-2\eta)} \right)^4 \right) = \tilde{O}\left( \frac{s}{(1-2\eta)^4} \mathrm{polylog}\left( d, \frac{1}{\epsilon}, \frac{1}{\delta} \right) \right).
\end{aligned}
$$

The proof is complete. $\qquad\square$

# C  Proof of Theorem 5

*Proof.* We first observe that if REFINE is run for $T$ iterations with bandwidth $b$, then with high probability, it will encounter $O\left( \frac{T}{b} \right)$ unlabeled examples. This is because, $O\left( \frac{1}{b} \right)$ calls of EX suffices to obtain an example that lies in $B_{\hat{w}_t, b}$, since it has probability mass $\Omega(b)$ (see Lemma 38).

For the initialization step (line 2), Algorithm 1 first draws $O\left( \frac{s \ln d}{(1-2\eta)^2} \right)$ unlabeled examples from $D_X$; then it runs REFINE with $\tilde{O}\left( \frac{s}{(1-2\eta)^4} \cdot \mathrm{polylog}\,(d) \right)$ iterations with bandwidth $b = \Theta\left( (1-2\eta)^2 \right)$. Therefore, this step queries $\tilde{O}\left( \frac{s}{(1-2\eta)^6} \cdot \mathrm{polylog}\,(d) \right)$ times to EX.

Now we discuss the number of unlabeled examples in phases 1 through $k_0$. For the $k$-th phase, Algorithm 1 runs REFINE with $\tilde{O}\left( \frac{s}{(1-2\eta)^2} \cdot \mathrm{polylog}\,(d) \right)$ iterations with bandwidth $b = \Theta((1-2\eta)2^{-k})$, which encounters $\tilde{O}\left( \frac{s \cdot 2^k}{(1-2\eta)^3} \cdot \mathrm{polylog}\,(d) \right)$ examples. Therefore, summing over $k =$

$1, 2, \ldots, k_0$, the total number of unlabeled examples queried to EX is $\tilde{O}\left(\frac{s \cdot 2^{k_0}}{(1-2\eta)^3} \cdot \text{polylog}(d)\right) = \tilde{O}\left(\frac{s}{(1-2\eta)^3 \epsilon} \cdot \text{polylog}(d)\right)$.

Summing over the two parts, the total number of queries to the unlabeled example oracle EX is $\tilde{O}\left(\frac{s}{(1-2\eta)^3} \cdot \left(\frac{1}{(1-2\eta)^3} + \frac{1}{\epsilon}\right) \cdot \text{polylog}(d)\right)$. $\qquad\square$

## D  Analysis of local convergence: Proof of Theorem 4

Before delving into the proof of Theorem 4, we first introduce an useful definition. Recall that $\hat{w}$ is the $\ell_2$-normalized vector of $w$. Define function

$$f_{u,b}(w) \overset{\text{def}}{=} \mathbb{E}_{(x,y)\sim D_{\hat{w},b}}\left[\,|u \cdot x| \cdot \mathbf{1}(\text{sign}(w \cdot x) \neq \text{sign}(u \cdot x))\right]. \tag{1}$$

Note that for any $l > 0$ and $w$ in $\mathbb{R}^d$, $f_{u,b}(w) = f_{u,b}(lw)$; specifically, $f_{u,b}(w) = f_{u,b}(\hat{w})$. We will discuss the structure of $f_{u,b}$ in detail in Appendix F; roughly speaking, $f_{u,b}(w)$ provides a "distance measure" between $w$ and $u$.

The lemma below motivates the above definition of $f_{u,b}$.

**Lemma 6.** *Given a vector $w_t$ and an example $(x_t, y_t)$ sampled randomly from $D_{\hat{w}_t, b}$, define $\hat{y}_t = \text{sign}(w_t \cdot x_t)$. Define the gradient vector induced by this example as $g_t = (-\frac{1}{2}y_t + (\frac{1}{2} - \eta)\hat{y}_t)x_t$. Then,*

$$\mathbb{E}_{x_t, y_t \sim D_{\hat{w}_t, b}}\left[\langle u, -g_t\rangle\right] \geq (1 - 2\eta)f_{u,b}(w_t). \tag{2}$$

*Proof.* Throughout this proof, we will abbreviate $\mathbb{E}_{x_t, y_t \sim D_{\hat{w}_t, b}}$ as $\mathbb{E}$. By the definition of $g_t$, we have

$$\mathbb{E}\left[\langle u, -g_t\rangle\right] = \mathbb{E}\left[\frac{1}{2}y_t\langle u, x_t\rangle - \left(\frac{1}{2} - \eta\right)\hat{y}_t\langle u, x_t\rangle\right].$$

We first look at $\mathbb{E}\left[\frac{1}{2}y_t\langle u, x_t\rangle\right]$. Observe that

$$\mathbb{E}\left[\frac{1}{2}y_t\langle u, x_t\rangle\right] = \mathbb{E}\left[\frac{1}{2}\mathbb{E}[y_t \mid x_t]\langle u, x_t\rangle\right] \geq \mathbb{E}\left[\frac{1}{2}|\langle u, x_t\rangle|(1 - 2\eta)\right]$$

where the equality uses the tower property of conditional expectation, and the inequality uses Lemma 7 below.

Therefore, by linearity of expectation, along with the above inequality, we have:

$$\mathbb{E}\left[\frac{1}{2}y_t\langle u, x_t\rangle - \left(\frac{1}{2} - \eta\right)\hat{y}_t\langle u, x_t\rangle\right]$$
$$\geq \left(\frac{1}{2} - \eta\right)\mathbb{E}\left[\,|\langle u, x_t\rangle|\,(1 - \text{sign}(\langle u, x_t\rangle)\text{sign}(\langle w, x_t\rangle))\right]$$
$$= (1 - 2\eta)\mathbb{E}\left[\,|\langle u, x\rangle|\,\mathbf{1}(\text{sign}(\langle w, x\rangle) \neq \text{sign}(\langle u, x\rangle))\right] = (1 - 2\eta)f_{u,b}(w).$$

The lemma follows. $\qquad\square$

**Lemma 7.** *Fix any $x \in \mathcal{X}$. Suppose $y$ is drawn from $D_{Y|X=x}$ that satisfies the $\eta$-bounded noise assumption with respect to $u$. Then,*

$$\langle u, x\rangle\,\mathbb{E}[y \mid x] \geq (1 - 2\eta)\,|\langle u, x\rangle|.$$

*Proof.* We do a case analysis. If $\langle u, x\rangle \geq 0$, by Assumption 1, $\mathbb{P}(Y = 1|X = x) \geq 1 - \eta$, making $\mathbb{E}[y \mid x] = \mathbb{P}(Y = 1|X = x) - \mathbb{P}(Y = -1|X = x) \geq (1 - 2\eta)$; symmetrically, if $\langle u, x\rangle < 0$, $\mathbb{E}[y \mid x] \leq -(1 - 2\eta)$. In summary, $\langle u, x\rangle\,\mathbb{E}[y \mid x] \geq (1 - 2\eta)\,|\langle u, x\rangle|$. $\qquad\square$

We have the following general lemma that provides a characterization of the iterates $\{w_t\}_{t=1}^T$ produced by Algorithm 2.

**Lemma 8.** *There exists an absolute constant $c > 0$ such that the following holds. Suppose we are given a vector $w_1$ in $\mathbb{R}^d$, convex set $\mathcal{K}$, and scalars $r_1, r_2 > 0$ such that:*

1. *$\|w_1 - u\|_1 \leq r_1$;*

2. *Both $w_1$ and $u$ are in $\mathcal{K}$;*

3. *For all $w$ in $\mathcal{K}$, $\|w - u\|_2 \leq r_2$; in addition, for all $w$ in $\mathcal{K}$, $\|w\|_2 \leq 1$.*

*If Algorithm 2 is run with initialization $w_1$, step size $\alpha > 0$, bandwidth $b \in [0, \frac{\pi}{72}]$, constraint set $\mathcal{K}$, regularizer $R(w) = \Phi_{w_1}(w)$, number of iterations $T$, then, with probability $1 - \delta$,*

$$\frac{1}{T} \sum_{t=1}^{T} f_{u,b}(w_t) \leq c \cdot \left( \frac{\alpha \left( \ln \frac{Td}{\delta b} \right)^2}{(1 - 2\eta)} + \frac{r_1^2 \ln d}{\alpha(1 - 2\eta)T} + \frac{b}{(1 - 2\eta)} + \frac{(b + r_2)}{(1 - 2\eta)} \left( \sqrt{\frac{\ln \frac{1}{\delta}}{T}} + \frac{\ln \frac{1}{\delta}}{T} \right) \right).$$

The proof of this lemma is rather technical; we defer it to the end of this section.

We now give an application of this lemma towards our proof of Theorem 4.

**Corollary 9.** *Suppose we are given an $s$-sparse unit vector $v$ such that $\|v - u\|_2 \leq 2\theta$, where $\theta \leq \frac{\pi}{32}$. If Algorithm 2 is run with initializer $v$, bandwidth $b = \Theta\left((1 - 2\eta)\theta\right)$, step size $\alpha = \Theta\left((1 - 2\eta)\theta / \ln^2(\frac{d}{\delta'\theta(1-2\eta)})\right)$, constraint set $\mathcal{K} = \{w : \|w\|_2 \leq 1, \|w - v\|_2 \leq 2\theta\}$, regularizer $R(w) = \Phi_v(w)$, number of iterations $T = O\left(\frac{s}{(1-2\eta)^2}(\ln \frac{d}{\delta'\theta(1-2\eta)})^3\right)$, then, with probability $1 - \delta'$,*

$$\frac{1}{T} \sum_{t=1}^{T} f_{u,b}(w_t) \leq \frac{\theta}{50 \cdot 3^4 \cdot 2^{33}}.$$

*Proof.* We first check that the premises of Lemma 8 are satisfied with $w_1 = v$, $r_1 = \sqrt{8s}\theta$ and $r_2 = 4\theta$. To see this, observe that:

1. As both $v$ and $u$ are $s$-sparse, their difference $v - u$ is $2s$-sparse. Therefore, $\|v - u\|_1 \leq \sqrt{2s}\|v - u\|_2 \leq \sqrt{8s}\theta$;

2. Both $u$ and $w$ are unit vectors, and have $\ell_2$ distance at most $2\theta$ to $v$, therefore they are both in $\mathcal{K}$;

3. For all $w$ in $\mathcal{K}$, $\|w - u\|_2 \leq \|w - v\| + \|v - u\|_2 \leq 4\theta$. Moreover, every $w$ in $\mathcal{K}$ satisfies the constraint $\|w\|_2 \leq 1$ by the definition of $\mathcal{K}$.

Therefore, applying Lemma 8 with our choice of $r_1$, $r_2$, $\alpha$, $b$, and $T$, we have that, the following four terms: $\alpha\left(\ln \frac{Td}{\delta b}\right)^2 / (1 - 2\eta)$, $r_1^2 \ln d / \alpha(1 - 2\eta)T$, $b/(1 - 2\eta)$, $(b + r_2)/(1 - 2\eta)\left(\sqrt{\ln \frac{1}{\delta'}/T} + \ln \frac{1}{\delta'}/T\right)$, are all at most $\frac{\theta}{c \cdot 50 \cdot 3^4 \cdot 2^{35}}$. Consequently,

$$\frac{1}{T} \sum_{t=1}^{T} f_{u,b}(w_t) \leq c \cdot 4 \cdot \frac{\theta}{c \cdot 50 \cdot 3^4 \cdot 2^{35}} \leq \frac{\theta}{50 \cdot 3^4 \cdot 2^{33}}.$$

The proof is complete. $\qquad\square$

We also need the following useful claim.

**Claim 10.** *If $\theta(w, u) \leq \frac{\pi}{2}$, and $f_{u,b}(w) \leq \frac{\theta}{5 \cdot 3^4 \cdot 2^{21}}$, then $\theta(w, u) \leq \frac{\theta}{5}$.*

*Proof.* We conduct a case analysis:

1. If $\theta(w, u) \leq 36b$, we are done, because from our choice of $b$, $36b \leq \frac{\theta}{5}$.

2. Otherwise, $\theta(w, u) \in [36b, \frac{\pi}{2}]$. In this case, by item 1 of Lemma 22 in Appendix F, we have that $f_{u,b}(w) \geq \frac{\theta(w,u)}{3^4 \cdot 2^{21}}$. In conjunction with the premise that $f_{u,b}(w) \leq \frac{\theta}{5 \cdot 3^4 \cdot 2^{21}}$, we get that $\theta(w, u) \leq \frac{\theta}{5}$.

In summary, in both cases, we have $\theta(w, u) \leq \frac{\theta}{5}$. □

*Proof of Theorem 4.* First, given a unit vector $\tilde{v}$ such that $\theta(\tilde{v}, u) \leq \theta$, we have that $\|\tilde{v} - u\|_2 = 2\sin\frac{\theta(\tilde{v},u)}{2} \leq \theta$. As $u$ is $s$-sparse, and $v = \mathcal{H}_s(\tilde{v})$, by Lemma 26, we have that $\|v - u\| \leq 2\theta$.

Next, by the definition of $\mathcal{K}$, for all $t$, $\|w_t - u\| \leq r_2 = 4\theta$. By Lemma 28, this implies that $\theta(w_t, u) \leq \pi \cdot 4\theta \leq 16\theta$. Moreover, by the fact that $\theta \leq \frac{\pi}{32}$, for all $t$, $\theta(w_t, u) \leq \frac{\pi}{2}$.

Now, applying Corollary 9, we have that with probability $1-\delta'$, the $\{w_t\}_{t=1}^T$ generated by Algorithm 2 are such that

$$\frac{1}{T}\sum_{t=1}^T f_{u,b}(w_t) \leq \frac{\theta}{50 \cdot 3^4 \cdot 2^{33}}.$$

Define $A = \{t \in [T] : f_{u,b}(w_t) \geq \frac{\theta}{5 \cdot 3^4 \cdot 2^{21}}\}$. As $\frac{1}{T}\sum_{t=1}^T f_{u,b}(w_t) \geq \frac{\theta}{5 \cdot 3^4 \cdot 2^{21}} \cdot \frac{1}{T}\sum_{t=1}^T \mathbf{1}(t \in A) = \frac{\theta}{5 \cdot 3^4 \cdot 2^{21}} \frac{|A|}{T}$, we have $\frac{|A|}{T} \leq \frac{5 \cdot 3^4 \cdot 2^{21}}{50 \cdot 3^4 \cdot 2^{33}} = \frac{1}{10 \cdot 2^{12}}$. Therefore, $\frac{|\bar{A}|}{T} \geq 1 - \frac{1}{10 \cdot 2^{12}}$, and for all $t \in \bar{A}$ we have $f_{u,b}(w_t) \leq \frac{\theta}{50 \cdot 3^4 \cdot 2^{21}}$; by Claim 10 above, we have $\theta(w_t, u) \leq \frac{\theta}{5}$ for these $t$.

Using the fact that for all $t$ in $A$, $\theta(w_t, u) \leq 16\theta$, and the fact that for all $t$ in $\bar{A}$, $\theta(w_t, u) \leq \frac{\theta}{5}$, we have:

$$\begin{aligned}
\frac{1}{T}\sum_{t=1}^T \cos\theta(w_t, u) &\geq \cos\frac{\theta}{5} \cdot \left(1 - \frac{1}{20 \cdot 2^{12}}\right) + \cos(16\theta) \cdot \frac{1}{20 \cdot 2^{12}} \\
&\geq \left(1 - \frac{\theta^2}{40}\right)\left(1 - \frac{1}{20 \cdot 2^{12}}\right) + \left(1 - \frac{(16\theta)^2}{2}\right)\frac{1}{20 \cdot 2^{12}} \\
&\geq 1 - \frac{\theta^2}{40} - \frac{\theta^2}{40} = 1 - \frac{\theta^2}{20} \geq \cos\frac{\theta}{2}.
\end{aligned}$$

where the second inequality uses item 2 of Lemma 23, the third inequality is by algebra, and the last inequality uses item 1 of Lemma 23.

The above inequality, in combination with Lemma 24 yields the following guarantee for $\tilde{w}$:

$$\cos\theta(\tilde{w}, u) \geq \frac{1}{T}\sum_{t=1}^T \cos\theta(w_t, u) \geq \cos\frac{\theta}{2}.$$

This implies that $\theta(\tilde{v}', u) \leq \frac{\theta}{2}$ since we set $\tilde{v}' = \tilde{w}$. □

### D.1  Proof of Lemma 8

Throughout this section, we define a filtration $\{\mathcal{F}_t\}_{t=0}^T$ as follows: $\mathcal{F}_0 = \sigma(w_1)$,
$$\mathcal{F}_t = \sigma(w_1, x_1, y_1, \ldots, w_t, x_t, y_t, w_{t+1}),$$
for all $t \in [T]$. As a shorthand, we write $\mathbb{E}_{t-1}[\cdot]$ for $\mathbb{E}[\cdot \mid \mathcal{F}_{t-1}]$.

*Proof of Lemma 8.* From standard analysis of online mirror descent [see e.g. 62, Theorem 6.8] with step size $\alpha$, constraint set $\mathcal{K}$ and regularizer $\Phi(w) = \frac{1}{2(p-1)}\|w - w_1\|_p^2$, we have that for every $u'$ in $\mathcal{K}$,

$$\alpha \cdot \left[\sum_{t=1}^T \langle w_t, g_t\rangle + \sum_{t=1}^T \langle -u', g_t\rangle\right] \leq D_\Phi(u', w_1) - D_\Phi(u, w_{T+1}) + \sum_{t=1}^T \alpha^2 \|g_t\|_q^2.$$

Let $u' = u$ in the above inequality, drop the negative term on the right hand side, and observe that $\|g_t\|_q \leq 2\|g_t\|_\infty \leq 2\|x_t\|_\infty$ (see Lemma 25), we have

$$\alpha \cdot \left[\sum_{t=1}^T \langle w_t, g_t\rangle + \sum_{t=1}^T \langle -u, g_t\rangle\right] \leq D_\Phi(u, w_1) + \sum_{t=1}^T 4\alpha^2 \|x_t\|_\infty^2.$$

Moving the first term to the right hand side, and divide both sides by $\alpha$, we get:

$$\sum_{t=1}^{T} \langle -u, g_t \rangle \leq \frac{D_\Phi(u, w_1)}{\alpha} + \sum_{t=1}^{T} \langle -w_t, g_t \rangle + 4\alpha \sum_{t=1}^{T} \|x_t\|_\infty^2. \tag{3}$$

Let us look at each of the terms closely. First, we can easily upper bound $D_\Phi(u, w_1)$ by assumption:

$$D_\Phi(u, w_1) = \frac{\|u - w_1\|_p^2}{2(p-1)} \leq \frac{\ln(8d) - 1}{2} r_1^2 \leq \frac{r_1^2 \ln(8d)}{2}. \tag{4}$$

where the first inequality uses the observation that as $p \geq 1$, $\|u - w_1\|_p^2 \leq \|u - w_1\|_1^2 \leq r_1^2$.

Let $W_t \stackrel{\text{def}}{=} \langle -w_t, g_t \rangle$. First, example $x_t$ is sampled from region $B_{\hat{w}_t, b}$, $|\langle \hat{w}_t, x_t \rangle| \leq b$. Moreover, by the assumption that $\mathcal{K} \subset \{w : \|w\|_2 \leq 1\}$, we have $\|w_t\|_2 \leq 1$, implying that $|\langle w_t, x_t \rangle| \leq b$. Therefore, $|W_t| = \left|\frac{1}{2}y_t - (\frac{1}{2} - \eta)\hat{y}_t\right| |\langle w_t, x_t \rangle| \leq b$. Consequently,

$$\sum_{t=1}^{T} W_t \leq T \cdot b. \tag{5}$$

Define $U_t \stackrel{\text{def}}{=} \langle -u, g_t \rangle$. By Lemma 6, $\mathbb{E}_{t-1} U_t \geq (1 - 2\eta) f_{u,b}(w_t)$. Moreover, Lemma 11 implies that there is a numerical constant $c_1 > 0$, such that with probability $1 - \delta/3$: $\left|\sum_{t=1}^{T} U_t - \mathbb{E}_{t-1} U_t\right| \leq c_1(b + r_2) \left(\sqrt{T \ln \frac{1}{\delta}} + \ln \frac{1}{\delta}\right)$. Consequently,

$$\sum_{t=1}^{T} (1 - 2\eta) f_{u,b}(w_t) \leq \sum_{t=1}^{T} \mathbb{E}_{t-1} U_t \leq \sum_{t=1}^{T} U_t + c_1(b + r_2) \left(\sqrt{T \ln \frac{1}{\delta}} + \ln \frac{1}{\delta}\right). \tag{6}$$

Moreover, by Lemma 13, there exists a constant $c_2 > 0$, such that with probability $1 - \delta/3$,

$$\sum_{t=1}^{T} \|x_t\|_\infty^2 \leq c_2 T \cdot \left(\ln \frac{Td}{\delta b}\right)^2. \tag{7}$$

Combining Equations (3), (4), (5), (6) and (7), along with union bound, we get that there exists a constant $c_3 > 0$, such that with probability $1 - \delta$:

$$(1 - 2\eta) \sum_{t=1}^{T} f_{u,b}(w_t) \leq c_3 \left(\alpha T \left(\ln \frac{Td}{\delta b}\right)^2 + \frac{r_1^2 \ln d}{\alpha} + bT + (b + r_2) \left(\sqrt{T \ln \frac{1}{\delta}} + \ln \frac{1}{\delta}\right)\right).$$

The theorem follows by dividing both sides by $(1 - 2\eta)T$. □

**Lemma 11.** *Recall that $U_t = \langle u, -g_t \rangle$. There is a numerical constant $c$ such that the following holds. We have that with probability $1 - \delta$,*

$$\left|\sum_{t=1}^{T} (U_t - \mathbb{E}_{t-1} U_t)\right| \leq c(b + r_2) \left(\sqrt{T \ln \frac{1}{\delta}} + \ln \frac{1}{\delta}\right). \tag{8}$$

*Proof.* By item 3 of the premise of Lemma 8, along with the fact that $w_t \in \mathcal{K}$, $\|u - w_t\| \leq r_2$, we hence have $\|u - \hat{w}_t\| \leq 2r_2$ using Lemma 27. Therefore, Lemma 12 implies the existence of constants $\beta$ and $\beta'$ such that for all $a \geq 0$,

$$\mathbb{P}_{x_t \sim D_{\hat{w}_t, b}}(|u \cdot x_t| \geq a) \leq \beta \exp\left(-\beta' \frac{a}{r_2 + b}\right).$$

Let $M_t = (\frac{1}{2}y_t - (\frac{1}{2} - \eta)\hat{y}_t)$. Observe that $|M_t| \leq 1$. Therefore, $U_t = \langle u, g_t \rangle = M_t u \cdot x_t$ has the exact same tail probability bound, i.e.

$$\mathbb{P}_{x_t \sim D_{\hat{w}_t, b}}(|U_t| \geq a) \leq \beta \exp\left(-\beta' \frac{a}{r_2 + b}\right).$$

The lemma now follows from Lemma 36 in Appendix H with the setting of $Z_t = U_t$. □

Lemma 11 relies on the following useful lemma from [5].

**Lemma 12.** *There exist numerical constants $\beta$ and $\beta'$ such that for any isotropic log-concave distribution $D_X$ over $\mathbb{R}^d$, any unit vector $\hat{w}$ in $\mathbb{R}^d$ and $u \in \mathbb{R}^d$ with $\|u\|_2 \leq 1$, $\|u - \hat{w}\| \leq r$, any scalar $b$ in $[0, 1]$, the following holds for all $a \geq 0$:*

$$\mathbb{P}_{x \sim D_{\hat{w},b}} (|u \cdot x| \geq a) \leq \beta \exp\left(-\beta' \frac{a}{r+b}\right).$$

*Proof.* Using Lemma 3.3 of [5] with $C = 1$, we have that there exists numerical constants $c_0, c'_0 > 0$, such that for any $K \geq 4$,

$$\mathbb{P}_{x \sim D_{w,b}} \left(|u \cdot x| \geq K\sqrt{r^2 + b^2}\right) \leq c \exp\left(-c'_0 K \sqrt{1 + \frac{b^2}{r^2}}\right) \leq c_0 \exp\left(-c'_0 K\right).$$

Therefore, for every $a \geq 4(r + b) \geq 4\sqrt{r^2 + b^2}$,

$$\mathbb{P}_{x \sim D_{w,b}} (|u \cdot x| \geq a) \leq c_0 \exp\left(-c'_0 \frac{a}{\sqrt{r^2 + b^2}}\right) \leq c_0 \exp\left(-c'_0 \frac{a}{(r+b)}\right).$$

In addition, for every $a < 4(r + b)$, $\mathbb{P}_{x \sim D_{w,b}} (|u \cdot x| \geq a) \leq 1$ trivially holds, in which case,

$$\mathbb{P}_{x \sim D_{w,b}} (|u \cdot x| \geq a) \leq 1 \leq \exp(4c'_0) \exp\left(-c'_0 \frac{a}{(r+b)}\right).$$

Therefore, we can find new numerical constants $\beta = \max(c_0, \exp(4c'_0))$ and $\beta' = c'_0$, such that

$$\mathbb{P}_{x \sim D_{w,b}} (|u \cdot x| \geq a) \leq \beta \exp\left(-\beta' \frac{a}{r+b}\right)$$

holds. $\square$

The lemma below provides a coarse bound on the last term in the regret guarantee (3).

**Lemma 13.** *With probability $1 - \delta$, $\sum_{t=1}^{T} \|x_t\|_\infty^2 \leq T \cdot \left(17 + \ln \frac{Td}{\delta b}\right)^2$.*

*Proof.* Given $x \in \mathbb{R}^d$ and $j \in [d]$, let $x^{(j)}$ be the $j$-th coordinate of $x$. As $D_X$ is isotropic log-concave, for $x \sim D_X$, from Lemma 39 we have that for all coordinates $j$ in $\{1, \ldots, d\}$ and every $a > 0$,

$$\mathbb{P}_{x \sim D_X} \left(\left|x^{(j)}\right| \geq a\right) \leq \exp(-a + 1). \tag{9}$$

Therefore, using union bound, we have

$$\mathbb{P}_{x \sim D_X} (\|x\|_\infty \geq a) \leq d \exp(-a + 1).$$

In addition, as $b \in [0, \frac{\pi}{72}] \subset [0, \frac{1}{9}]$, we have by Lemma 37, $\mathbb{P}_{x \sim D_X} (x \in R_{\hat{w},b}) \geq \frac{b}{2^{16}}$.

Now, by the simple fact that $\mathbb{P}(A|B) \leq \frac{\mathbb{P}(A)}{\mathbb{P}(B)}$, we have that

$$\mathbb{P}_{x \sim D_{\hat{w},b}} (\|x\|_\infty \geq a) = \mathbb{P}_{x \sim D_X} (\|x\|_\infty \geq a | x \in R_{\hat{w},b}) \leq \frac{\mathbb{P}_{x \sim D_X} (\|x\|_\infty \geq a)}{\mathbb{P}_{x \sim D_X} (x \in R_{\hat{w},b})}$$

$$\leq \frac{2^{16} d}{b} \exp(-a + 1).$$

Therefore, taking $a = 17 + \ln \frac{Td}{\delta b}$ in the above inequality, we get that, the above event happens with probability at most $\frac{\delta}{T}$. In other words, with probability $1 - \frac{\delta}{T}$,

$$\|x_t\|_\infty \leq 17 + \ln \frac{Td}{\delta b}. \tag{10}$$

Thus, taking a union bound, we get that with probability $1 - \delta$, for every $t$, Equation (10) holds. As a result, $\sum_{t=1}^{T} \|x_t\|_\infty^2 \leq T \cdot \left(17 + \ln \frac{Td}{\delta b}\right)^2$. $\square$

# E  Analysis of initialization: Proof of Theorem 3

## E.1  Obtaining a halfspace constraint on $u$

Before going into the proof of Theorem 3, we introduce a few notations. Throughout this section, we use $\mathbb{E}$ to denote $\mathbb{E}_{(x,y)\sim D}$ as a shorthand. Denote by $\bar{w} \stackrel{\text{def}}{=} \mathbb{E}[xy]$, and denote $\hat{\mathbb{E}}$ as the empirical expectation over $(x_i, y_i)_{i=1}^m$; with this notation, $w_{\text{avg}} = \hat{\mathbb{E}}[xy]$. Denote by $w_{\tilde{s}} = \mathcal{H}_{\tilde{s}}(w_{\text{avg}})$; in this notation, $w^\sharp = \frac{w_{\tilde{s}}}{\|w_{\tilde{s}}\|}$.

**Lemma 14.** *If Algorithm 3 is run with hard-thresholding parameter $\tilde{s} = 81 \cdot 2^{38} \cdot \frac{s}{(1-2\eta)^2}$, number of labeled examples $m = 81 \cdot 2^{51} \cdot \frac{s}{(1-2\eta)^2} \ln \frac{8d}{\delta'}$, then with probability $1 - \delta'/2$, the unit vector $w^\sharp$ obtained at line 3 is such that*

$$\left\langle w^\sharp, u \right\rangle \geq \frac{(1-2\eta)}{9 \cdot 2^{19}}. \tag{11}$$

*Proof.* First, Lemma 15 below implies that

$$\langle \bar{w}, u \rangle \geq \frac{(1-2\eta)}{9 \cdot 2^{16}}. \tag{12}$$

Moreover, as $u$ is a unit vector, and $D_X$ is isotropic log-concave, $\langle u, x \rangle$ comes from a one-dimensional isotropic log-concave distribution. In addition, $y$ is a random variable that takes values in $\{\pm 1\}$. Therefore, by Lemma 34, $y\langle u, x \rangle$ is $(32, 16)$-subexponential. Lemma 31, in allusion to the choice of $m$, implies that with probability $1 - \delta'/4$,

$$\left| \frac{1}{m} \sum_{i=1}^m [y_i \langle u, x_i \rangle] - \mathbb{E}[y \langle u, x \rangle] \right| \leq 32 \sqrt{\frac{2 \ln \frac{8}{\delta}}{m}} + 32 \frac{\ln \frac{8}{\delta}}{m} \leq \frac{(1-2\eta)}{9 \cdot 2^{17}} \tag{13}$$

Thus,

$$\langle w_{\text{avg}}, u \rangle = \left\langle \frac{1}{m} \sum_{i=1}^m y_i x_i, u \right\rangle \geq \frac{(1-2\eta)}{9 \cdot 2^{16}} - \frac{(1-2\eta)}{9 \cdot 2^{17}} \geq \frac{(1-2\eta)}{9 \cdot 2^{17}}. \tag{14}$$

Now, consider $w_{\tilde{s}} = \mathcal{H}_{\tilde{s}}(w_{\text{avg}})$. By Lemma 17 shown below, with the choice of $m$, we have that with probability $1 - \delta'/4$, $\|w_{\tilde{s}}\|_2 \leq 2$. Hence, by union bound, with probability $1 - \delta'/2$, both Equation (14) and $\|w_{\tilde{s}}\|_2 \leq 2$ hold.

In this event, Lemma 16 (also shown below), in combination with the fact that $\tilde{s} = \frac{81 \cdot 2^{38} s}{(1-2\eta)^2}$, implies that

$$\langle w_{\tilde{s}}, u \rangle \geq \langle w_{\text{avg}}, u \rangle - \sqrt{\frac{s}{\tilde{s}}} \|w_{\tilde{s}}\| \geq \frac{(1-2\eta)}{9 \cdot 2^{17}} - \frac{(1-2\eta)}{9 \cdot 2^{19}} \cdot 2 = \frac{(1-2\eta)}{9 \cdot 2^{18}}. \tag{15}$$

By the fact that $w^\sharp = \frac{w_{\tilde{s}}}{\|w_{\tilde{s}}\|}$ and using again $\|w_{\tilde{s}}\| \leq 2$, we have

$$\left\langle w^\sharp, u \right\rangle \geq \frac{1}{2} \langle w_{\tilde{s}}, u \rangle \geq \frac{(1-2\eta)}{9 \cdot 2^{19}},$$

which is the desired result. $\square$

**Lemma 15.** *Suppose that Assumption 1 is satisfied. Then*

$$\langle \bar{w}, u \rangle \geq \frac{(1-2\eta)}{9 \cdot 2^{16}}.$$

*Proof.* Recall that $\bar{w} = \mathbb{E}[xy]$. We have

$$
\begin{aligned}
\langle \bar{w}, u \rangle &= \mathbb{E}[y(u \cdot x)] \\
&= \mathbb{E}\left[\mathbb{E}[y \mid x](u \cdot x)\right] \\
&\geq (1-2\eta)\mathbb{E}[|u \cdot x|] \geq \frac{(1-2\eta)}{9 \cdot 2^{16}},
\end{aligned}
$$

where the first equality is by the linearity of inner product and expectation, the second equality is by the tower property of conditional expectation. The first inequality uses Lemma 7. For the last inequality, we use the fact that $z = u \cdot x$ can be seen as drawn from a one-dimensional isotropic log-concave distribution with density $f_Z$, along with Lemma 37 with $d = 1$ with states that for every $z \in [0, 1/9]$, $f_Z(z) \geq 2^{-16}$, making $\mathbb{E}_{z \sim f_Z}[|z|]$ bounded from below by $\frac{1}{9 \cdot 2^{16}}$. $\qquad\square$

The following lemma is inspired by Lemma 12 of [87].

**Lemma 16.** *For any vector $a$ and any $s$-sparse unit vector $u$, we have*

$$|\langle \mathcal{H}_{\tilde{s}}(a), u \rangle - \langle a, u \rangle| \leq \sqrt{\frac{s}{\tilde{s}}} \|\mathcal{H}_{\tilde{s}}(a)\|.$$

*Proof.* Let $\Omega$ be the support of $\mathcal{H}_{\tilde{s}}(a)$, and $\Omega'$ be the support of $u$. Given any vector $v$, denote by $v_1$ (resp. $v_2, v_3$) the vector obtained by zeroing out all elements outside $\Omega \setminus \Omega'$ (resp. $\Omega \cap \Omega'$, $\Omega' \setminus \Omega$) from $v$. With this notation, it can be seen that $\mathcal{H}_{\tilde{s}}(a) = a_2 + a_3$, $\langle \mathcal{H}_k(a), u \rangle = \langle a_2, u_2 \rangle$, $\langle a, u \rangle = \langle a_2, u_2 \rangle + \langle a_3, u_3 \rangle$. Thus, it suffices to prove that $|\langle a_3, u_3 \rangle| \leq \sqrt{\frac{s}{\tilde{s}}} \|\mathcal{H}_{\tilde{s}}(a)\|$.

First, this holds in the trivial case that $a_3$ is a zero-vector. Now suppose that $a_3$ is non-zero. By the definition of $\mathcal{H}_{\tilde{s}}$, this implies that all the elements of $\mathcal{H}_{\tilde{s}}(a)$ is non-zero, and hence $\|\mathcal{H}_{\tilde{s}}(a)\|_0 = \tilde{s}$. In addition, every element of $a_3$ has absolute value smaller than that of $\mathcal{H}_{\tilde{s}}(a)$. Consequently, the average squared element of $a_3$ is larger than that of $\mathcal{H}_{\tilde{s}}(a)$, namely

$$\frac{\|a_3\|^2}{\|a_3\|_0} \leq \frac{\|\mathcal{H}_{\tilde{s}}(a)\|^2}{\|\mathcal{H}_{\tilde{s}}(a)\|_0}. \tag{16}$$

Since $\|a_3\|_0 = |\Omega' \setminus \Omega| \leq |\Omega'| = s$, and $\|\mathcal{H}_{\tilde{s}}(a)\|_0 = \tilde{s}$, we obtain $\|a_3\| \leq \sqrt{\frac{s}{\tilde{s}}} \|a_1\|$. The result follows by observing that $|\langle a_3, u_3 \rangle| \leq \|a_3\| \cdot \|u_3\| \leq \|a_3\|$ where the first inequality is by Cauchy-Schwarz and the second one is from the premise that $\|u\| = 1$. $\qquad\square$

Recall that $w_{\text{avg}} = \hat{\mathbb{E}}[xy]$ is the vector obtained by empirical average all $x_i y_i$'s. In the lemma below, we argue that the $\ell_2$ norm of $w_{\tilde{s}} = \mathcal{H}_{\tilde{s}}(w_{\text{avg}})$ is small. As a matter of fact, we show a stronger result that, keeping any $\tilde{s}$ elements of vector $w$ (and zeroing out the rest) makes the resulting vector have a small norm.

**Lemma 17.** *Suppose $\tilde{s} \in [d]$ is a natural number. With probability $1 - \delta'/4$ over the draw of $m = 2^{13} \cdot \tilde{s} \ln \frac{8d}{\delta'}$ examples, the following holds: For any subset $\Omega \subset [d]$ of size $\tilde{s}$, we have that $\|(w_{\text{avg}})_\Omega\| \leq 2$, where $(w_{\text{avg}})_\Omega$ is obtained by zeroing out all but the elements in $\Omega$.*

*Proof.* We prove the lemma in two steps: first, we show that $\bar{w} = \mathbb{E}[xy]$ must have a small $\ell_2$ norm – specifically, this implies that $\|\bar{w}_\Omega\|_2$ is small; second, we show that $\bar{w}$ and $w_{\text{avg}}$ are close to each other entrywise. Then we combine these two observations to show that $(w_{\text{avg}})_\Omega$ has a small $\ell_2$ norm. Write the vector $\bar{w} = (\bar{w}^{(1)}, \bar{w}^{(2)}, \ldots, \bar{w}^{(d)})$ and the vector $x = (x^{(1)}, x^{(2)}, \ldots, x^{(d)})$.

For the first step, by Lemma 18 shown below, we have

$$\sum_{i \in \Omega} \left(\bar{w}^{(j)}\right)^2 \leq \sum_{j=1}^{d} \left(\bar{w}^{(j)}\right)^2 = \sum_{j=1}^{d} (\mathbb{E}\left[x^{(j)} y\right])^2 \leq 1. \tag{17}$$

For the second step, we know that as $x^{(j)}$ is drawn from an isotropic log-concave and $y$ take values in $\{\pm 1\}$, by Lemma 34 in Appendix H, $x^{(j)} y$ is $(32, 16)$-subexponential. Therefore, by Lemma 31, along with union bound, we have that with probability $1 - \delta$, for all coordinates $j$ in $[d]$,

$$\left| w_{\text{avg}}^{(j)} - \bar{w}^{(j)} \right| = \left| \hat{\mathbb{E}}[x^{(j)} y] - \mathbb{E}[x^{(j)} y] \right| \leq 32 \sqrt{2 \frac{\ln \frac{2d}{\delta}}{m}} + 32 \frac{\ln \frac{2d}{\delta}}{m} \leq \frac{1}{\sqrt{\tilde{s}}}, \tag{18}$$

where the last inequality is from our setting of $m$.

The above two items together imply that,

$$\sum_{j \in \Omega} \left(w_{\text{avg}}^{(j)}\right)^2 \leq \sum_{j \in \Omega} 2 \left(\bar{w}^{(j)}\right)^2 + 2 \left(w_{\text{avg}}^{(j)} - \bar{w}^{(j)}\right)^2 \leq 2 + 2 \frac{\tilde{s}}{\tilde{s}} \leq 4. \tag{19}$$

The lemma is concluded by recognizing that the left hand side is $\|(w_{\mathrm{avg}})_\Omega\|^2$ and by setting $\delta = \delta'/4$ in (18). $\qquad \square$

**Lemma 18.** *Given a vector $x \in \mathcal{X}$, we write $x = (x^{(1)}, x^{(2)}, \dots, x^{(d)})$. We have*

$$\sum_{j=1}^d \left(\mathbb{E}\big[x^{(j)} y\big]\right)^2 \le 1. \tag{20}$$

*Proof.* Denote by function $\zeta(x) \overset{\text{def}}{=} \mathbb{E}[y|x]$. As $y \in \{\pm 1\}$, we have that for every $x$, $\zeta(x) \in [-1, +1]$. In this notation, by the tower property of expectation, $\mathbb{E}\big[x^{(j)} y\big] = \mathbb{E}\big[x^{(j)} \zeta(x)\big]$.

For $f, g$ in $L^2(D_X)$, we denote by $\langle f, g \rangle_{L^2(D_X)} = \mathbb{E}_{x \sim D_X}[f(x)g(x)]$ their inner product in $L^2(D_X)$. As $D_X$ is isotropic,

$$\left\langle x^{(j)}, x^{(j)} \right\rangle_{L^2(D_X)} = \mathbb{E}_{x \sim D_X}\left[x^{(j)} x^{(j)}\right] = \begin{cases} 1, & i = j, \\ 0, & i \ne j. \end{cases}$$

Therefore, $x^{(1)}, \dots, x^{(d)}$ is a set of orthonormal functions in $L^2(D_X)$. This implies

$$\sum_{j=1}^d \left(\mathbb{E}\left[x^{(j)} \zeta(x)\right]\right)^2 = \sum_{j=1}^d \left\langle \zeta, x^{(j)} \right\rangle_{L^2(D_X)}^2 \le \langle \zeta, \zeta \rangle_{L^2(D_X)} \le 1. \tag{21}$$

where the equality is from the definition of $\langle f, g \rangle_{L^2(D_X)}$, the first inequality is from Bessel's inequality, and the second inequality uses the fact that $\zeta(x)^2 \in [0, 1]$ and $D_X$ is a probability measure. This completes the proof. $\qquad \square$

### E.2 Obtaining a vector that has a small angle with $u$

One technical challenge in directly applying the same analysis of Theorem 4 to the initialization phase is that, some of the $w_t$'s obtained may have large obtuse angles with $u$ (e.g. $\theta(w_t, u)$ is close to $\pi$), making their corresponding $f_{u,b}(w_t)$ value small. To prevent this undesirable behavior, Algorithm 3 add a linear constraint $\langle w, w^\sharp \rangle \ge \frac{(1-2\eta)}{9 \cdot 2^{19}}$ on the set $\mathcal{K}$ when applying REFINE, which ensures that all vectors in $\mathcal{K}$ will have angle with $u$ bounded away from $\pi$. The lemma below formalizes this intuition.

Recall that Algorithm 3 sets $\mathcal{K} = \left\{ w : \|w\|_2 \le 1, \|w\|_1 \le \sqrt{s}, \langle w, w^\sharp \rangle \ge \frac{(1-2\eta)}{9 \cdot 2^{19}} \right\}$.

**Lemma 19.** *For any two vectors $w_1, w_2 \in \mathcal{K}$, the angle between them, $\theta(w_1, w_2)$, is such that*

$$\theta(w_1, w_2) \le \pi - \frac{(1-2\eta)}{9 \cdot 2^{19}}.$$

*Proof.* First, by the definition of $\mathcal{K}$, for $w_1, w_2$ in $\mathcal{K}$, we have $\langle w_i, w^\sharp \rangle \ge \frac{(1-2\eta)}{9 \cdot 2^{19}}$ for $i = 1, 2$. In addition, by the definition of $\mathcal{K}$, both $w_1$ and $w_2$ have norms at most 1. This implies that their normalized version, $\hat{w}_1$ and $\hat{w}_2$, satisfies, $\langle \hat{w}_i, w^\sharp \rangle \ge \frac{(1-2\eta)}{9 \cdot 2^{19}}$ for $i = 1, 2$.

For $i = 1, 2$, let $\hat{w}_i = \hat{w}_{i,\|} + \hat{w}_{i,\perp}$ be an orthogonal decomposition, where $\hat{w}_{i,\|}$ (resp. $\hat{w}_{i,\perp}$) denotes the component of $\hat{w}_i$ parallel to (resp. orthogonal to) $w^\sharp$. As $\|\hat{w}_i\| \le 1$, we have that $\|\hat{w}_{i,\perp}\| \le 1$, implying that $|\langle \hat{w}_{1,\perp}, \hat{w}_{2,\perp} \rangle| \le \|\hat{w}_{1,\perp}\| \cdot \|\hat{w}_{2,\perp}\| \le 1$. In addition, $\langle \hat{w}_{1,\|}, \hat{w}_{2,\|} \rangle = \langle \hat{w}_1, w^\sharp \rangle \cdot \langle \hat{w}_2, w^\sharp \rangle \ge \left( \frac{(1-2\eta)}{9 \cdot 2^{19}} \right)^2$. Therefore,

$$\cos\theta(w_1, w_2) = \langle \hat{w}_1, \hat{w}_2 \rangle = \langle \hat{w}_{1,\|}, \hat{w}_{2,\|} \rangle + \langle \hat{w}_{1,\perp}, \hat{w}_{2,\perp} \rangle \ge -1 + \left( \frac{(1-2\eta)}{9 \cdot 2^{19}} \right)^2.$$

By item 3 of Lemma 23, we get that

$$-1 + \frac{1}{2} (\theta(w_1, w_2) - \pi)^2 \ge -1 + \left( \frac{(1-2\eta)}{9 \cdot 2^{19}} \right)^2,$$

The above inequality, in combination with the basic fact that $\theta(w_1, w_2) \in [0, \pi]$, implies that $\theta(w_1, w_2) \le \pi - \frac{(1-2\eta)}{9 \cdot 2^{19}}$. $\qquad \square$

The following lemma is the main result of this subsection, which shows that by using the new constraint set $\mathcal{K}$ in Algorithm 3, REFINE obtains a vector with constant angle with $u$ with $O\left(\frac{s}{(1-\eta)^4}\right)$ labels.

**Lemma 20.** *Suppose we are given a unit vector $w^\sharp$ such that $\langle w^\sharp, u \rangle \geq \frac{(1-2\eta)}{9 \cdot 2^{19}}$. If Algorithm 2 is run with initialization $w_1 = 0$, bandwidth $b = \Theta\left((1-2\eta)^2\right)$, step size $\alpha = \Theta\left((1-2\eta)^2 / \left(\ln \frac{d}{\delta'(1-2\eta)}\right)^2\right)$, constraint set $\mathcal{K} = \left\{ w : \|w\|_2 \leq 1, \langle w, w^\sharp \rangle \geq \frac{(1-2\eta)}{9 \cdot 2^{19}} \right\}$, regularizer $\Phi(w) = \frac{1}{2(p-1)} \|w\|_p^2$, number of iterations $T = O\left(\frac{s}{(1-2\eta)^4} \left(\ln \frac{d}{\delta'(1-2\eta)}\right)^3\right)$, then with probability $1 - \frac{\delta'}{2}$, it returns a vector $\tilde{v}_0$ such that $\theta(\tilde{v}_0, u) \leq \frac{\pi}{32}$.*

*Proof.* We first check the premises of Lemma 8 with the chosen $w_1 \in \mathcal{K}$, constraint set

$$\mathcal{K} = \left\{ w : \|w\|_2 \leq 1, \|w\|_1 \leq \sqrt{s}, \langle w, w^\sharp \rangle \geq \frac{(1-2\eta)}{9 \cdot 2^{19}} \right\},$$

$r_1 = 2\sqrt{s}, r_2 = 2$:

1. Observe that $\|u\|_1 \leq \sqrt{\|u\|_0} \|u\|_2 \leq \sqrt{s}$; in addition, by the definition of $\mathcal{K}$, $\|w_1\|_1 \leq \sqrt{s}$. Therefore, $\|w_1 - u\|_1 \leq \|u\|_1 + \|w_1\|_1 \leq 2\sqrt{s} = r_1$;

2. $w_1$ is in $\mathcal{K}$ by definition; for $u$, we have $\|u\|_2 = 1$ by definition; $\|u\|_1 \leq \sqrt{s}$ by the argument above; $\langle u, w^\sharp \rangle \geq \frac{(1-2\eta)}{9 \cdot 2^{19}}$. Therefore, $u$ is also in $\mathcal{K}$.

3. For every $w$ in $\mathcal{K}$, as $\|w\|_2 \leq 1$, we have $\|w - u\| \leq \|w\|_2 + \|u\|_2 = r_2$; in addition, by the definition of $\mathcal{K}$, every $w$ in $\mathcal{K}$ satisfies that $\|w\| \leq 1$.

Therefore, applying Lemma 8, we have that with probability $1 - \frac{\delta'}{2}$,

$$\frac{1}{T} \sum_{t=1}^{T} f_{u,b}(w_t) \leq c \cdot \left( \frac{\alpha \left(\ln \frac{2Td}{\delta' b}\right)^2}{(1-2\eta)} + \frac{4s \ln d}{\alpha(1-2\eta)T} + \frac{b}{(1-2\eta)} + \frac{(b+2)}{(1-2\eta)} \left( \sqrt{\frac{\ln \frac{1}{\delta'}}{T}} + \frac{\ln \frac{1}{\delta'}}{T} \right) \right).$$

Specifically, with the choice of $\alpha = \Theta\left((1-2\eta)^2 / \left(\ln \frac{d}{\delta'(1-2\eta)}\right)^2\right)$, $b = O\left((1-2\eta)^2\right)$, $T = O\left(\frac{s}{(1-2\eta)^4} \left(\ln \frac{d}{\delta'(1-2\eta)}\right)^3\right)$, we have that all four terms $\alpha\left(\ln \frac{2Td}{\delta' b}\right)^2 / (1-2\eta)$, $\frac{4s \ln d}{\alpha(1-2\eta)T}$, $b/(1-2\eta)$, $(b+2) \cdot \left( \sqrt{\frac{\ln \frac{1}{\delta'}}{T}} + \frac{\ln \frac{1}{\delta'}}{T} \right) / (1-2\eta)$ are all at most $\frac{(1-2\eta)}{c \cdot 5 \cdot 3^6 \cdot 2^{51}}$, implying that

$$\frac{1}{T} \sum_{t=1}^{T} f_{u,b}(w_t) \leq 4c \cdot \frac{(1-2\eta)}{c \cdot 5 \cdot 3^6 \cdot 2^{51}} \leq \frac{(1-2\eta)}{5 \cdot 3^6 \cdot 2^{49}}.$$

Define $A = \left\{ t \in [T] : f_{u,b}(w_t) \geq \frac{(1-2\eta)}{3^6 \cdot 2^{40}} \right\}$. As $\frac{1}{T} \sum_{t=1}^{T} f_{u,b}(w_t) \geq \frac{(1-2\eta)}{3^6 \cdot 2^{40}} \cdot \frac{1}{T} \sum_{t=1}^{T} \mathbf{1}(t \in A) = \frac{(1-2\eta)}{3^6 \cdot 2^{40}} \frac{|A|}{T}$, we have $\frac{|A|}{T} \leq \frac{1}{5 \cdot 2^9}$. Therefore, $\frac{|\bar{A}|}{T} \geq 1 - \frac{1}{5 \cdot 2^9}$, and for every $t$ in $\bar{A}$, $w_t$ is such that $f_{u,b}(w_t) < \frac{(1-2\eta)}{3^6 \cdot 2^{40}}$.

We establish the following claim that characterizes the iterates $w_t$ where $t \in \bar{A}$.

**Claim 21.** *If $w \in \mathcal{K}$ and $f_{u,b}(w) < \frac{(1-2\eta)}{3^6 \cdot 2^{40}}$, then $\theta(w, u) < \frac{(1-2\eta)}{9 \cdot 2^{19}}$.*

*Proof.* First, we show that it is impossible for $\theta(w, u) \geq \frac{\pi}{2}$. By Lemma 19, for all $w$ in $\mathcal{K}$, we have that $\theta(w, u) \leq \pi - \frac{(1-2\eta)}{9 \cdot 2^{19}}$. By the choice of $b$, we know that $\theta(w, u) \leq \pi - 36b$. By item 2 of Lemma 22, we have

$$f_{u,b}(w) \geq \frac{\pi - \theta(w, u)}{3^4 \cdot 2^{21}} \geq \frac{(1-2\eta)}{3^6 \cdot 2^{40}}, \tag{22}$$

which contradicts with the premise that $f_{u,b}(w) < \frac{(1-2\eta)}{3^6 \cdot 2^{40}}$.

Therefore, $\theta(w,u) \in [0, \frac{\pi}{2}]$. We now conduct a case analysis.

1. If $\theta(w,u) \leq 36b$, then by the definition of $b$, we automatically have $\theta(w,u) < \frac{(1-2\eta)}{9 \cdot 2^{19}}$.

2. Otherwise, $\theta(w,u) \in [36b, \frac{\pi}{2}]$. In this case, by item 1 of Lemma 22, we have

$$f_{u,b}(w) \geq \frac{\theta(w,u)}{3^4 \cdot 2^{21}}.$$

   This inequality, in conjunction with the assumption that $f_{u,b}(w) < \frac{(1-2\eta)}{3^6 \cdot 2^{40}}$, implies that $\theta(w,u) \leq \frac{(1-2\eta)}{9 \cdot 2^{19}}$.

In summary, in both cases, we have $\theta(w,u) \leq \frac{(1-2\eta)}{9 \cdot 2^{19}}$. This completes the proof. $\qquad\square$

Claim 21 above implies that, for all $t$ in $\bar{A}$, $\theta(w_t,u) \leq \frac{(1-2\eta)}{9 \cdot 2^{19}} \leq \frac{\pi}{128}$. In addition, $\frac{|\bar{A}|}{T} \geq 1 - \frac{1}{5 \cdot 2^9}$. Combining the above facts with the simple fact that $\cos\theta(w_t,w) \geq -1$ for all $t$ in $A$, we have:

$$
\begin{aligned}
\frac{1}{T} \sum_{t=1}^{T} \cos(\theta_t, u) &\geq \cos\frac{\pi}{128} \cdot \left(1 - \frac{1}{5 \cdot 2^9}\right) - 1 \cdot \left(\frac{1}{5 \cdot 2^9}\right) \\
&\geq \left(1 - \frac{1}{2}\left(\frac{\pi}{128}\right)^2\right) \cdot \left(1 - \frac{1}{5 \cdot 2^9}\right) - \frac{1}{5 \cdot 2^9} \\
&\geq 1 - \frac{1}{5}\left(\frac{\pi}{32}\right)^2 \\
&\geq \cos\frac{\pi}{32}
\end{aligned}
$$

where the first inequality is from the above conditions on $A$ and $\bar{A}$ we obtained; the second inequality uses item 2 of Lemma 23; the third inequality is by algebra; the last inequality uses item 1 of Lemma 23.

Combining the above result with Lemma 24, we have the following for $\tilde{v}_0 = \tilde{w}$:

$$\cos\theta(\tilde{v}_0, u) = \cos\theta(\tilde{w}, u) \geq \frac{1}{T}\sum_{t=1}^{T}\cos(\theta_t, u) \geq \cos\frac{\pi}{32}.$$

This implies that $\theta(\tilde{v}_0, u) \leq \frac{\pi}{32}$. $\qquad\square$

Theorem 3 is now a direct consequence of Lemmas 14 and 20.

*Proof of Theorem 3.* First, by Lemma 14, we have that there exists an event $E_1$ that happens with probability $1 - \delta'/2$, in which the unit vector $w^\sharp$ obtained is such that $\langle w^\sharp, u \rangle \geq \frac{(1-2\eta)}{9 \cdot 2^{19}}$. In addition, Lemma 20 states that there exists an event $E_2$ with probability $1 - \delta'/2$, in which if $\langle w^\sharp, u \rangle \geq \frac{(1-2\eta)}{9 \cdot 2^{19}}$, it returns $\tilde{v}_0$ such that $\theta(\tilde{v}_0, u) \leq \frac{\pi}{32}$. The theorem follows from considering the event $E_1 \cap E_2$, which happens with probability $1 - \delta'$, in which $\tilde{v}_0$, the final output of Algorithm 3, satisfies that $\theta(\tilde{v}_0, u) \leq \frac{\pi}{32}$. The total number of label queries made by Algorithm 3 is:

$$n = O\left(\frac{s \ln d}{(1-2\eta)^2} + \frac{s}{(1-2\eta)^4}\left(\ln\frac{d}{\delta'(1-2\eta)}\right)^3\right) = O\left(\frac{s}{(1-2\eta)^4}\left(\ln\frac{d}{\delta'(1-2\eta)}\right)^3\right).$$

$\qquad\square$

# F  The structure of function $f_{u,b}$

Recall that
$$f_{u,b}(w) = \mathbb{E}_{(x,y) \sim D_{\hat{w},b}} \left[ |u \cdot x| \, \mathbf{1}(\text{sign}\,(\langle w, x \rangle) \neq \text{sign}\,(\langle u, x \rangle)) \right].$$
Note that for all $w$, $f_{u,b}(w) \geq 0$. In this section, we show a few key properties of $f_{u,b}$, in that if $w$ has an acute angle with $u$, $f_{u,b}(w)$ behaves similar to the $\theta(w, u)$; if $w$ has an obtuse angle with $u$, $f_{u,b}(w)$ behaves similar to $\pi - \theta(w, u)$.

**Lemma 22.** *Suppose $w$ and $u$ are two unit vectors; in addition, suppose $b \leq \frac{\pi}{72}$. We have:*

1. *If $\theta(u, w) \in [36b, \frac{\pi}{2}]$, then $f_{u,b}(w) \geq \frac{\theta(w,u)}{3^4 \cdot 2^{21}}$.*

2. *If $\theta(u, w) \in [\frac{\pi}{2}, \pi - 36b]$, then $f_{u,b}(w) \geq \frac{\pi - \theta(w,u)}{3^4 \cdot 2^{21}}$.*

*Proof.* We prove the two items respectively.

1. For the first item, we denote by $\phi \stackrel{\text{def}}{=} \theta(u, w)$. Define region
$$R_1 = \left\{ x : \langle w, x \rangle \in [0, b], \langle u, x \rangle \in \left[ -\frac{\sin \phi}{36}, -\frac{\sin \phi}{18} \right] \right\}.$$

It can be easily seen that $R_1$ is a subset of the disagreement region between $w$ and $u$. In other words,
$$\mathbf{1}(x \in R_1) \leq \mathbf{1}(\text{sign}\,(\langle w, x \rangle) \neq \text{sign}\,(\langle u, x \rangle)).$$

It suffices to show that, region $R_1$ has probability mass at least $\frac{b}{9 \cdot 2^{18}}$ wrt $D_X$. To see why it completes the proof, observe that
$$\begin{aligned}
&\mathbb{E}_{x \sim D_X} \left[ |u \cdot x| \, \mathbf{1}(|\langle w, x \rangle| \leq b) \mathbf{1}(\text{sign}\,(\langle w, x \rangle) \neq \text{sign}\,(\langle u, x \rangle)) \right] \\
&\geq \mathbb{E}_{x \sim D_X} \left[ |u \cdot x| \, \mathbf{1}(x \in R_1) \right] \\
&\geq \frac{\sin \phi}{36} \cdot \mathbb{E}_{x \sim D_X} \mathbf{1}(x \in R_1) \\
&\geq \frac{\phi}{72} \cdot \mathbb{P}_{x \sim D_X} (x \in R_1) \geq \frac{\phi \cdot b}{3^4 \cdot 2^{21}},
\end{aligned}$$
where the first inequality uses the fact that $R_1$ is a subset of both $\{x : |\langle w, x \rangle| \leq b\}$ and $\{x : \text{sign}\,(\langle w, x \rangle) \neq \text{sign}\,(\langle u, x \rangle)\}$; the second inequality uses the fact that for all $x$ in $R_1$, $|u \cdot x| \geq \frac{\sin \phi}{36}$; the third inequality uses the elementary fact that $\sin \phi \geq \frac{\phi}{2}$.

As $\mathbb{P}_{x \sim D_X} (|\langle w, x \rangle| \leq b) \leq b$ by Lemma 38, this implies that
$$f_{u,b}(w) = \frac{\mathbb{E}_{x \sim D_X} \left[ |u \cdot x| \, \mathbf{1}(|\langle w, x \rangle| \leq b) \mathbf{1}(\text{sign}\,(\langle w, x \rangle) \neq \text{sign}\,(\langle u, x \rangle)) \right]}{\mathbb{P}_{x \sim D_X} (|\langle w, x \rangle| \leq b)} \geq \frac{\phi \cdot b}{9 \cdot 2^{18} \cdot b}$$
$$= \frac{\phi}{9 \cdot 2^{18}}.$$

Now we turn to lower bounding the probability mass of $R_1$ wrt $D_X$. We first project $x$ down to the subspace spanned by $\{w, u\}$ - call the projected value $z = (z_1, z_2) \in \mathbb{R}^2$. Observe that $z$ can also be seen as drawn from an isotropic log-concave distribution in $\mathbb{R}^2$; denote by $f_Z$ its probability density function.

Without loss of generality, suppose $w = (0, 1)$ and $u = (\sin \phi, \cos \phi)$. It can be now seen that $x \in R_1$ iff $z$ lies in the parallelogram $ABDC$, denoted as $\tilde{R}_1$, where $A = (\frac{1}{36} + \frac{b}{\tan \phi}, b)$, $B = (\frac{1}{18} + \frac{b}{\tan \phi}, b)$, $C = (\frac{1}{36}, 0)$, $D = (\frac{1}{18}, 0)$. See Figure 1 for an illustration. Crucially, $\|\overline{OC}\| = \|\overline{CD}\| = \frac{1}{36}$, $\|\overline{AC}\| = \|\overline{BD}\| = \frac{b}{\sin \phi} \leq \frac{1}{18}$, as $b \leq \frac{\phi}{36} \leq \frac{\sin \phi}{18}$. Therefore, by triangle inequality, all four vectices, $A, B, C, D$ have distance at most $\frac{1}{9}$ to the origin. Therefore, for all $z \in \tilde{R}_1$, $\|z\| \leq \frac{1}{9}$. By Lemma 37, this implies that $f_Z(z) \geq 2^{-16}$ for all $z$ in $\tilde{R}_1$. Moreover, the area of parallelogram $\tilde{R}_1$ is equal to $b \cdot \frac{1}{36} = \frac{b}{36}$.

Therefore,

$$\mathbb{P}_{x \sim D_X}\left(x \in R_1\right) = \mathbb{P}_{z \sim D_Z}\left(z \in \tilde{R}_1\right) = \int_{\tilde{R}_1} f_Z(z)dz \geq 2^{-16} \cdot \frac{b}{36} = \frac{b}{9 \cdot 2^{18}}.$$

This completes the proof of the claim.

2. The proof of the second item uses similar lines of reasoning as the first. We denote by $\phi \overset{\text{def}}{=} \pi - \theta(u,w)$. Define region

$$R_2 = \left\{ x : \langle w, x \rangle \in [-b, 0], \langle u, x \rangle \in \left[\frac{\sin\phi}{36}, \frac{\sin\phi}{18}\right] \right\}.$$

It can be easily seen that $R_2$ is a subset of the disagreement region between $w$ and $u$. In other words,

$$\mathbf{1}(x \in R_2) \leq \mathbf{1}(\text{sign}\left(\langle w, x \rangle\right) \neq \text{sign}\left(\langle u, x \rangle\right)).$$

It suffices to show that, region $R_2$ has probability mass at least $\frac{b}{9 \cdot 2^{18}}$ wrt $D_X$. To see why it completes the proof, observe that

$$\begin{aligned}
&\mathbb{E}_{x \sim D_X}\left[|u \cdot x|\, \mathbf{1}(|\langle w, x \rangle| \leq b)\mathbf{1}(\text{sign}\left(\langle w, x \rangle\right) \neq \text{sign}\left(\langle u, x \rangle\right))\right] \\
&\geq \mathbb{E}_{x \sim D_X}\left[|u \cdot x|\, \mathbf{1}(x \in R_2)\right] \\
&\geq \frac{\sin\phi}{36} \cdot \mathbb{E}_{x \sim D_X}\mathbf{1}(x \in R_2) \\
&= \frac{\phi}{72}\mathbb{P}_{x \sim D_X}\left(x \in R_2\right) \geq \frac{b \cdot \phi}{3^4 \cdot 2^{21}},
\end{aligned}$$

where the first inequality uses the fact that $R_2$ is a subset of both $\{x : |\langle w, x \rangle| \leq b\}$ and $\{x : \text{sign}\left(\langle w, x \rangle\right) \neq \text{sign}\left(\langle u, x \rangle\right)\}$; the second inequality uses the fact that for all $x$ in $R_2$, $|u \cdot x| \geq \frac{\sin\phi}{36}$; the third inequality uses the elementary fact that $\sin\phi \geq \frac{\phi}{2}$.

As $\mathbb{P}_{x \sim D_X}\left(|\langle w, x \rangle| \leq b\right) \leq b$ by Lemma 38, this implies that

$$\begin{aligned}
f_{u,b}(w) = \frac{\mathbb{E}_{x \sim D_X}\left[|u \cdot x|\, \mathbf{1}(|\langle w, x \rangle| \leq b)\mathbf{1}(\text{sign}\left(\langle w, x \rangle\right) \neq \text{sign}\left(\langle u, x \rangle\right))\right]}{\mathbb{P}_{x \sim D_X}\left(|\langle w, x \rangle| \leq b\right)} &\geq \frac{\phi \cdot b}{3^4 \cdot 2^{21} \cdot b} \\
&= \frac{\phi}{3^4 \cdot 2^{21}}.
\end{aligned}$$

Now we lower bound the probability mass of $R_2$ wrt $D_X$. We first project $x$ down to the subspace spanned by $\{w, u\}$ - call the projected value $z = (z_1, z_2) \in \mathbb{R}^2$. Observe that $z$ can also be seen as drawn from an isotropic log-concave distribution on $\mathbb{R}^2$; denote by its density $f_Z(z)$.

Without loss of generality, suppose $w = (0, 1)$ and $u = (\sin\phi, -\cos\phi)$. It can be now seen that $x \in R_2$ iff $z$ lies in the parallelogram $CDBA$, denoted as $\tilde{R}_2$, where $A = (\frac{1}{36} - \frac{b}{\tan\phi}, -b)$, $B = (\frac{1}{18} - \frac{b}{\tan\phi}, -b)$, $C = (\frac{1}{36}, 0)$, $D = (\frac{1}{18}, 0)$. See Figure 2 for an illustration. Crucially, $\|\overline{OC}\| = \|\overline{CD}\| = \frac{1}{36}$, $\|\overline{AC}\| = \|\overline{BD}\| = \frac{b}{\sin\phi} \leq \frac{1}{18}$, as $b \leq \frac{\phi}{36} \leq \frac{\sin\phi}{18}$. Therefore, by triangle inequality, all four vertices $A, B, C, D$ have distance at most $\frac{1}{9}$ to the origin. Therefore, for all $z \in \tilde{R}_1$, $\|z\| \leq \frac{1}{9}$. This implies that $f_Z(z) \geq 2^{-16}$ for all $z$ in $\tilde{R}_2$. Moreover, the area of parallelogram $\tilde{R}_2$ is equal to $b \cdot \frac{1}{36} = \frac{b}{36}$.

Therefore,

$$\mathbb{P}_{x \sim D_X}\left(x \in R_2\right) = \mathbb{P}_{z \sim D_Z}\left(z \in \tilde{R}_2\right) = \int_{\tilde{R}_2} f_Z(z)dz \geq 2^{-16} \cdot \frac{b}{36} = \frac{b}{9 \cdot 2^{18}}.$$

This completes the proof of the claim. $\qquad\square$

Figure 1: An illustration of parallelogram region $\tilde{R}_1$ (the shaded region). Its four boundaries are: lines $AB$ and $CD$, which are $\{z : \langle w, z \rangle = b\}$ and $\{z : \langle w, z \rangle = 0\}$; lines $AC$ and $BD$, which are $\left\{ z : \langle u, z \rangle = -\frac{\sin \phi}{36} \right\}$ and $\left\{ z : \langle u, z \rangle = -\frac{\sin \phi}{18} \right\}$ respectively.

Figure 2: An illustration of parallelogram region $\tilde{R}_2$ (the shaded region). Its four boundaries are: lines $AB$ and $CD$, which are $\{z : \langle w, z \rangle = -b\}$ and $\{z : \langle w, z \rangle = 0\}$; lines $AC$ and $BD$, which are $\left\{ z : \langle u, z \rangle = \frac{\sin \phi}{36} \right\}$ and $\left\{ z : \langle u, z \rangle = \frac{\sin \phi}{18} \right\}$ respectively.

# G  Basic inequalities

**Lemma 23.** *If $\theta \in [0, \pi]$, then:*

1. $\cos \theta \leq 1 - \frac{\theta^2}{5}$.

2. $\cos \theta \geq 1 - \frac{\theta^2}{2}$.

3. $\cos \theta \leq -1 + \frac{1}{2}(\theta - \pi)^2$.

4. $\cos \theta \geq -1 + \frac{1}{5}(\theta - \pi)^2$.

**Lemma 24** (Averaging effects on angle)**.** *Suppose we have a sequence of unit vectors $w_1, \ldots, w_T$. Let $\tilde{w} = \frac{1}{T} \sum_{t=1}^{T} w_t$ be their average. Suppose $\frac{1}{T} \sum_{t=1}^{T} \cos \theta(w_t, u) \geq 0$. Then, $\cos \theta(\tilde{w}, u) \geq \frac{1}{T} \sum_{t=1}^{T} \cos \theta(w_t, u)$.*

*Proof.* We note that

$$\langle \tilde{w}, u \rangle = \frac{1}{T} \sum_{t=1}^{T} \langle w_t, u \rangle = \frac{1}{T} \sum_{T=1}^{T} \cos \theta(w_T, u) \geq 0.$$

In addition, by the convexity of $\ell_2$ norm, $\|\tilde{w}\|_2 = \|\frac{1}{T}\sum_{t=1}^{T} w_t\|_2 \leq \frac{1}{T}\sum_{t=1}^{T}\|w_t\| \leq 1$. This implies that

$$\cos\theta(\tilde{w}, u) = \left\langle \frac{\tilde{w}}{\|\tilde{w}\|}, u \right\rangle \geq \langle \tilde{w}, u \rangle = \frac{1}{T}\sum_{t=1}^{T}\cos\theta(w_t, u).$$

$\square$

**Lemma 25.** *Recall that $q = \ln(8d)$. Then for every $x$ in $\mathbb{R}^d$, $\|x\|_q \leq 2\|x\|_\infty$.*

*Proof.* By algebra, $\|x\|_q = \left(\sum_{i=1}^{d}|x_i|^q\right)^{\frac{1}{q}} \leq (d\|x\|_\infty^q)^{\frac{1}{q}} \leq 2\|x\|_\infty$. $\square$

We need the following elementary lemmas in our proofs. See e.g. [88] for the proof.

**Lemma 26.** *If $v, u$ are two vectors in $\mathbb{R}^d$, and $u$ is $s$-sparse, then, $\|\mathcal{H}_s(v) - u\|_2 \leq 2\|v - u\|_2$.*

**Lemma 27.** *Suppose $v$ is a unit vector in $\mathbb{R}^d$. Then for any $w$ in $\mathbb{R}^d$, $\|\hat{w} - v\|_2 \leq 2\|w - v\|_2$.*

**Lemma 28.** *If $v$ is a unit vector in $\mathbb{R}^d$, and $w$ is a vector in $\mathbb{R}^d$, then $\theta(w, v) \leq \pi\|w - v\|_2$.*

## H  Probability tail bounds

In this section we present a few well-known results about concentrations of random variables and martingales that are instrumental in our proofs. We include the proofs of some of the results here because we would like to explicitly track dependencies on relevant parameters.

We start by recalling a few facts about subexponential random variables; see e.g. [83] for a more thorough treatment on this topic.

**Definition 29.** *A random variable $X$ with is called $(\sigma, b)$-subexponential, if for all $\lambda \in [-\frac{1}{b}, \frac{1}{b}]$,*

$$\mathbb{E}e^{\lambda(X - \mathbb{E}[X])} \leq e^{\frac{\sigma^2\lambda^2}{2}}. \tag{23}$$

**Lemma 30.** *Suppose $Z$ is $(\sigma, b)$-subexponential, then with probability $1 - \delta$,*

$$|Z - \mathbb{E}Z| \leq \sqrt{2\sigma^2\ln\frac{2}{\delta}} + 2b\ln\frac{2}{\delta}.$$

**Lemma 31.** *Suppose $X_1, \ldots, X_n$ are iid $(\sigma, b)$-subexponential random variables, then $\frac{1}{n}\sum_{i=1}^{n}X_i$ is $(\frac{\sigma}{\sqrt{n}}, \frac{b}{n})$-subexponential. Consequently, with probability $1 - \delta$,*

$$\left|\frac{1}{n}\sum_{i=1}^{n}(X_i - \mathbb{E}[X_i])\right| \leq \sqrt{\frac{2\sigma^2}{n}\ln\frac{2}{\delta}} + \frac{2b}{n}\ln\frac{2}{\delta}.$$

We next show the following fact: if a random variable has a subexponential tail probability, then it is subexponential.

**Lemma 32.** *Suppose $Z$ is a random variable such that $\mathbb{P}(|Z| > a) \leq C\exp\left(-\frac{a}{\sigma}\right)$ for some $C \geq 1$. Then,*

$$\mathbb{E}e^{\frac{|Z|}{2\sigma(\ln C + 1)}} \leq 4.$$

*Proof.* We bound the left hand side as follows:

$$\begin{aligned}
\mathbb{E}e^{\frac{|Z|}{2\sigma(\ln C + 1)}} &= \int_0^\infty \mathbb{P}\left(e^{\frac{|Z|}{2\sigma(\ln C + 1)}} \geq s\right)ds \\
&= \int_0^\infty \mathbb{P}\left(|Z| \geq 2\sigma(\ln C + 1)\ln s\right)ds \\
&\leq \int_0^\infty \min\left(1, \frac{C}{s^{2(\ln C + 1)}}\right)ds \\
&= \int_0^e \min\left(1, \frac{C}{s^{2(\ln C + 1)}}\right)ds + \int_e^\infty \min\left(1, \frac{C}{s^{2(\ln C + 1)}}\right)ds \\
&\leq e + \int_e^\infty Ce^{-2\ln C}s^{-2}ds \leq 4.
\end{aligned}$$

where the first equality is from a basic equality for nonnegative random variable $Y$: $\mathbb{E}[Y] = \int_0^\infty \mathbb{P}(Y \geq t)dt$; the second equality is by rewriting the event in terms of $|Z|$; the first inequality is from the assumption on $|Z|$'s tail probability and the simple fact that the probability of an event is always at most 1; the third equality is by decomposing the integration to integration on two intervals; the second inequality uses the fact that the first integral is at most $e$, and the integrand in the second integral is at most $Ce^{-2\ln C}s^{-2}$ as $s \geq e$; the last inequality uses the fact that $C \geq 1$ and $e + \frac{1}{e} \leq 4$. $\qquad\square$

**Lemma 33.** *For random variable $Z$ and some $\lambda_0 \in \mathbb{R}_+$, if $\mathbb{E}\exp(\lambda_0|Z|) \leq C_0$, then $Z$ is $(\frac{4\sqrt{C_0}}{\lambda_0}, \frac{4}{\lambda_0})$-subexponential.*

*Proof.* As $\mathbb{E}\exp(\lambda_0|Z|) = \sum_{i=0}^\infty \frac{\mathbb{E}|Z|^i \lambda_0^i}{i!}$, where each summand is an nonnegative number, we have that for all $i$,

$$\frac{\mathbb{E}|Z|^i \lambda_0^i}{i!} \leq \mathbb{E}\exp(\lambda_0|Z|) \leq C_0. \tag{24}$$

where the second inequality is by our assumption.

We introduce a new random variable $Z'$ such that $Z'$ has the exact same distribution as $Z$, and is independent of $Z$. Observe that $Z - Z'$ has a symmetric distribution, and therefore $\mathbb{E}(Z - Z')^i = 0$ for all odd $i$. We look closely at the moment generating function of $Z - Z'$:

$$\mathbb{E}\exp(\lambda(Z - Z')) = \sum_{i=0}^\infty \frac{\mathbb{E}(Z - Z')^i}{i!}\lambda^{2i} = \sum_{i=0}^\infty \frac{\mathbb{E}(Z - Z')^{2i}}{(2i)!}\lambda^{2i}$$

where the second equality uses the fact that $Z - Z'$ has a symmetric distribution. Importantly, by the conditional Jensen's Inequality and the convexity of exponential function, $\mathbb{E}\exp(\lambda(Z - \mathbb{E}[Z])) \leq \mathbb{E}\exp(\lambda(Z - Z'))$. Therefore, it suffices to bound $\mathbb{E}\exp(\lambda(Z - Z'))$ for all $\lambda \in [-\frac{\lambda_0}{4}, \frac{\lambda_0}{4}]$.

We have the following sequence of inequalities:

$$\begin{aligned}
\mathbb{E}\exp(\lambda(Z - Z')) &= \sum_{i=0}^\infty \frac{\mathbb{E}\left[|Z - Z'|^{2i}\right]\lambda^{2i}}{(2i)!} \\
&\leq 1 + \sum_{i=1}^\infty \frac{\mathbb{E}\left[|Z|^{2i}\right]2^{2i}\lambda_0^{2i}}{(2i)!} \cdot \left(\frac{\lambda}{\lambda_0}\right)^{2i} \\
&\leq 1 + C_0 \sum_{i=1}^\infty \left(\frac{2\lambda}{\lambda_0}\right)^{2i} \\
&\leq 1 + 2C_0 \left(\frac{2\lambda}{\lambda_0}\right)^2 \\
&\leq \exp\left(\frac{8C_0}{\lambda_0^2}\lambda^2\right).
\end{aligned}$$

where the first inequality we separate out the first constant term, and use the basic fact that $|z - z'|^j \leq 2^{j-1}(|z|^j + |z'|^j)$ for all $j \geq 1$, and the fact that $Z$ and $Z'$ has the same distribution; the second inequality uses Equation (24) that $\frac{\mathbb{E}|Z|^{2i}\lambda_0^{2i}}{(2i)!} \leq C_0$; the third inequality uses condition that $\left|\frac{\lambda}{\lambda_0}\right| \leq \frac{1}{4}$, and the elementary calculation that $\sum_{i=1}^\infty \left(\frac{\lambda}{\lambda_0}\right)^{2i} = \left(\frac{2\lambda}{\lambda_0}\right)^2 \cdot \frac{1}{1-(\frac{2\lambda}{\lambda_0})^2} \leq 8\left(\frac{\lambda}{\lambda_0}\right)^2$; the last inequality uses the simple fact that $1 + x \leq e^x$ for all $x$ in $\mathbb{R}$.

To conclude, we have that for all $\lambda \in [-\frac{\lambda_0}{4}, \frac{\lambda_0}{4}]$,

$$\mathbb{E}\exp(\lambda(Z - \mathbb{E}Z)) \leq \exp\left(\frac{8C_0}{\lambda_0^2}\lambda^2\right),$$

meaning that $Z$ is $\left(\frac{4\sqrt{C_0}}{\lambda_0}, \frac{4}{\lambda_0}\right)$-subexponential. $\qquad\square$

Importantly, based on the above two lemmas we have the following subexponential property of isotropic log-concave random variables.

**Lemma 34.** *If $X$ is a random variable drawn from a 1-dimensional isotropic log-concave distribution $D_X$, then $X$ is $(32, 16)$-subexponential. Moreover, for any random variable $Y$ such that $|Y| \leq 1$ almost surely, $YX$ is also $(32, 16)$-subexponential.*

*Proof.* By Lemma 39, we have that $\mathbb{P}(|X| \geq t) \leq e \cdot e^{-t}$. Applying Lemma 32 with $\sigma = 1$ and $C = e$, we have that $\mathbb{E}e^{\frac{|X|}{4}} \leq 4$. Now, using Lemma 33 with $\lambda_0 = \frac{1}{4}$ and $C_0 = 4$, we have that $X$ is $(32, 16)$-subexponential. The second statement follows from the exact same line of reasoning, starting from $\mathbb{P}(|YX| \geq t) \leq \mathbb{P}(|X| \geq t) \leq e \cdot e^{-t}$. $\square$

In the two lemmas below, we use the shorthand that $\mathbb{E}_t [\cdot] \stackrel{\text{def}}{=} \mathbb{E} [\cdot \mid \mathcal{F}_t]$, and $\mathbb{P}_t (\cdot) \stackrel{\text{def}}{=} \mathbb{P} (\cdot \mid \mathcal{F}_t)$.

We need the following standard martingale concentration lemma (see e.g. [85, Theorem 2.19]) where the conditional distribution of each martingale difference term has a subexponential distribution.

**Lemma 35.** *Suppose $\{Z_t\}_{t=1}^T$ is sequence of random variables adapted to filtration $\{\mathcal{F}_t\}_{t=1}^m$. In addition, each random variable $Z_t$ is conditionally $(\sigma, b)$-subexponential, formally,*

$$\mathbb{E}_{t-1} \left[ \exp \left( \lambda \left( Z_t - \mathbb{E}_{t-1} \left[ Z_t \right] \right) \right) \right] \leq \exp \left( \frac{\sigma^2 \lambda^2}{2} \right), \forall \lambda \in \left[ -\frac{1}{b}, \frac{1}{b} \right]. \tag{25}$$

*Then with probability $1 - \delta$,*

$$\left| \sum_{t=1}^T \left( Z_t - \mathbb{E}_{t-1} Z_t \right) \right| \leq \sigma \sqrt{2T \ln \frac{2}{\delta}} + 2b \ln \frac{2}{\delta}.$$

*Proof.* As all $Z_t$'s are conditionally $(\sigma, b)$-subexponential, Theorem 2.19 of [85] implies that $\sum_{t=1}^T \left( Z_t - \mathbb{E}_{t-1} Z_t \right)$ is $(\sigma\sqrt{T}, b)$-exponential, and for any $a > 0$,

$$\mathbb{P} \left( \left| \sum_{t=1}^T \left( Z_t - \mathbb{E}_{t-1} Z_t \right) \right| > a \right) \leq \max(2e^{-\frac{a^2}{2T\sigma^2}}, 2e^{-\frac{a}{2b}}).$$

Taking $a_0 = \max \left( \sqrt{2T \ln \frac{2}{\delta}}, 2b \ln \frac{2}{\delta} \right)$, we have $\mathbb{P} \left( \left| \sum_{t=1}^T \left( Z_t - \mathbb{E}_{t-1} Z_t \right) \right| > a_0 \right) \leq \delta$. The lemma is concluded by observing that $a_0 \leq \sqrt{2T \ln \frac{2}{\delta}} + 2b \ln \frac{2}{\delta}$. $\square$

Combining Lemmas 32, 33 and 35, we have the following useful inequality on the concentration of a martingale where each martingale difference has a subexponential probability tail. We note that Freedman's Inequality or Azuma-Hoeffding's Inequality does not directly apply, as they require the martingale difference to be almost surely bounded. A similar result for subgaussian martingale differences is shown in [70]; see also the discussions therein.

**Lemma 36.** *Suppose $\{Z_t\}_{t=1}^T$ is sequence of random variables adapted to filtration $\{\mathcal{F}_t\}_{t=1}^T$. For every $Z_t$, we have that $\mathbb{P}_{t-1}(|Z_t| > a) \leq C \exp \left( -\frac{a}{\sigma} \right)$ for some $C \geq 1$. Then, with probability $1 - \delta$,*

$$\left| \sum_{t=1}^T Z_t - \mathbb{E}_{t-1} Z_t \right| \leq 16\sigma(\ln C + 1) \left( \sqrt{2T \ln \frac{2}{\delta}} + \ln \frac{2}{\delta} \right).$$

*Proof.* First, by Lemma 32, we have that $\mathbb{E}_{t-1} \exp \left( \frac{|Z|}{2\sigma(\ln C+1)} \right) \leq 4$. Therefore, using Lemma 33, we have that $Z$ is $(16\sigma(\ln C + 1), 8\sigma(\ln C + 1))$-subexponential.

Therefore, by Lemma 35, we have that with probability $1 - \delta$,

$$\left| \sum_{t=1}^T Z_t - \mathbb{E}_{t-1} Z_t \right| \leq 16\sigma(\ln C + 1) \ln \frac{2}{\delta} + 16\sigma(\ln C + 1) \sqrt{2T \ln \frac{2}{\delta}}$$

$$\leq 16\sigma(\ln C + 1) \left( \sqrt{2T \ln \frac{2}{\delta}} + \ln \frac{2}{\delta} \right).$$

where the second inequality is by algebra. $\square$

# I   Basic facts about isotropic log-concave distributions

The following useful lemmas are from [57].

**Lemma 37.** *The statement below holds for $d = 1, 2$. Suppose $D_X$ is an isotropic log-concave distribution on $\mathbb{R}^d$, with probability density function $f$. Then, for all $x$ such that $\|x\|_2 \leq \frac{1}{9}$, $f(x) \geq 2^{-16}$.*

*Proof.* For any $d = 1, 2$, by items (a) and (d) of [57, Theorem 5.14], we have that for every $x$ such that $\|x\|_2 \leq \frac{1}{9}$, $f(x) \geq 2^{-9n\|x\|_2} f(0) \geq 2^{-d} f(0)$, and $f(0) \geq 2^{-7d}$. Therefore, for $x$ such that $\|x\|_2 \leq \frac{1}{9}$, $f(x) \geq 2^{-7d} \cdot 2^{-d} = 2^{-8d} \geq 2^{-16}$. $\qquad\square$

**Lemma 38.** *If $x$ is a random variable drawn from a 1-dimensional isotropic log-concave distribution, then for all $a, b \in \mathbb{R}$ such that $a < b$,*

$$\mathbb{P}(x \in [a, b]) \leq b - a.$$

**Lemma 39.** *If $x$ is a random variable drawn from a 1-dimensional isotropic log-concave distribution, then for every $t \geq 0$,*

$$\mathbb{P}(|x| > t) \leq e^{-t+1}.$$