[Reviews · NeurIPS 2020]

Review 1

Summary and Contributions: This paper gives the first algorithm for *active* learning halfspaces in the Massart noise model. It gets the "right" dependence on all the important parameters and works with respect to log-concave distributions.

Strengths: Seems like a very solid improvement on prior work.

Weaknesses: Doesn't clearly explain relationship to the *techniques* of Diakonikolas et al.

Correctness: Seems correct

Clarity: Reasonably well

Relation to Prior Work: Should discuss Diakonikolas et al. more in terms of techniques and the loss functions used.

Reproducibility: Yes

Additional Feedback:


Review 2

Summary and Contributions: The paper considers the active learning of halfspaces under the presence of Massart noise in the PAC setting. Under the assumption of an isotropic log concave data distribution and \eta bounded Massart noise, they present an algorithm that learns halfspaces to error \eps for any noise parameter in [0,1/2) and, has sample complexity polynomial in the dimension d, \log 1/\epsilon, and 1/(1 - 2\eta). When the Bayes classifier is promised to be an s-sparse halfspace, the sample complexity is polynomial in s, \log 1/\epsilon, and 1/(1 - 2\eta). In both cases these algorithms are the first to achieve a polynomial dependence on all the parameters of interest. The result is obtained using an approach inspired by regret minimization, a sequence of loss functions is constructed to capture the difference between the current vector and the underlying true halfspace. Making mirror descent updates with a suitably chosen regularizer, it can be shown that the regret gaurantee afforded by online optimization shows that the average normalized vector has a reduced angle from the true halfspace.

Strengths: The algorithms presented in the paper are major improvements over best known results for learning of halfspaces given an underlying isotropic logconcave distribution under Massart noise in 3 separate important areas: For active learning of general halfspaces, space halfspaces, and passive learning of halfspaces. In the final case the improvements are only polynomial bu the the dependence on dimension and the noise parameter are vastly reduced. The approach using regret minimization in this context is quite novel and elegant and raises the possibility for applying similar methods for other learning problems in this space. EDIT: My overall impression of the paper is unchanged after feedback, I think the approach and results are both instructive and valuable.

Weaknesses: While taking into account the page limits, it would be nice if there were some more intuition for why a mirror descent approach leads to such large improvements in this space compared to earlier methods.

Correctness: I did not find any errors in the proofs presented.

Clarity: The paper is well writtten, the problem is presented in a clear manner and the proofs are reasonably easy to follow.

Relation to Prior Work: Yes

Reproducibility: Yes

Additional Feedback:


Review 3

Summary and Contributions: This paper studies the problem of learning sparse halfspaces given access to a noisy point-label pair oracle. In particular, given underlying true halfspace h^*, the goal is to recover an \epsilon accurate + sparse representation h' of h^* using minimum number of noisy-oracle queries. The paper makes the following standard assumptions (i) the underlying distribution over the points is log-concave + isotropic (ii) The label noise model is Massart noise. Under these assumptions, the paper gives an efficient algorithm which \epsilon learns halfspaces using O(s/(1 - 2\eta)^4 \poly-log(d,\epsilon)) samples, making it the first linear in s-sample complexity algorithm in this setting. This is also known to be almost information theoretically optimal with the upper and lower bounds differing only by a factor of O(1/(1 - 2\eta)^2). The key idea behind the algorithm used in the paper is that active learning of sparse halfspaces can be phrased as a online convex optimization problem. Using this connection, the authors given an algorithm whose main component is a carefully designed mirror descent style update routine which gives multiplicatively better sparse approximates to the true halfspaces. Their full algorithm consists of two steps. 1. Initialize: In this step the algorithm finds a rough approximation to the true subspace by first averaging signed vectors constructed from sampled point label pairs, followed iterative mirror descent style updates with fixed learning rate. 2. Refine: This step involves mirror descent style updates with varying learning rates and feasible sets across epochs. Step 1. is where the sparsity assumption of the true halfspace is exploited; i.e., it allows for a rough s-sparse approximation of the true halfspace in \tilde{O}(s (1 - \eta)^{-4}) samples. In particular, this ensures that the diameter of feasible region for subsequent OMD style updates is at most \sqrt{s}, this primarily determines the rate of convergence for refinement routine. The OMD style updates in steps 1 and 2 requires a careful choice of update rule which ensures progress in terms of the corresponding bregman divergence. Furthermore, the feasible region for these updates are also carefully modified across iterations using a shrinking band which intuitively ensures that updates happen across most informative directions, thus ensuring fast covergence. The analysis of the algorithm combines known tools for learning halfspaces under log concave distribution, while addressing several additional technical challenges arising out of the specific choice of the update rule. Overall, I feel that this is an important result in the context of active learning of sparse halfspaces, and I recommend acceptance.

Strengths: > Makes a novel connection between active learning and the theory of online convex optimization which could be useful in the more broader context of active learning in continuous hypothesis classes. > The choice and the analysis of the update rule which also accounts for the label noise seems quite interesting.

Weaknesses: One weakness is that the sample complexity of the algorithm is worse by a factor of 1/(1 - 2\eta)^2 from the information theoretic lower bound

Correctness: I checked most of the proofs, the calculations seem correct.

Clarity: The paper is mostly well written, although the paper could do with a description and references for Online Mirror Descent.

Relation to Prior Work: Yes, the paper does a good job of describing previous related works, and connections to relevant literature.

Reproducibility: Yes

Additional Feedback: A couple of comments: > The algorithm probably needs an additional hard thresholding step before exiting, since thresholding is only done in the beginning of every iteration. > Although mirror descent is a common primitive in the convex optimization/online learning literature, it would be good to include a informal description of the framework (in the appendix) and some additional references for general audiences.

[Author Response · NeurIPS 2020]

We thank all reviewers for their constructive feedback.

**Techniques (Reviewer 2) and comparison to the concurrent work of [DKTZ20] (Reviewer 1):**

1. We propose to sequentially minimize a series of "potential functions" $f_{u,b}(w)$ (see Appendix F for definition)
for progressively smaller values of $b$. We believe that our time-varying setting of parameter $b$ in combina-
tion with this new potential function is the key to achieve the state-of-the-art $O\left(\mathrm{poly}(\frac{1}{1-2\eta}, \log\frac{1}{\epsilon})\right)$ label
complexity for active learning under Massart noise, which has not appeared before.

2. We also believe that our application of online linear optimization in (active) learning halfspaces is quite
novel: in order to optimize $f_{u,b}(w)$, we construct an online linear optimization problem whose *negative*
*benchmark performance* is an upper bound of $f_{u,b}(w)$ (Lemma 6)! To the best of our understanding, the
work of [DKTZ20] considers optimizing the global expectation of a sigmoid-like loss function (denoted as
$L(w)$), and its key observation is that the gradient of the expected loss $\nabla L(w)$ is large whenever $w$ and $u$
have a large angle. We are not sure if there is a connection there (although any connection would be very
interesting). To efficiently solve the online linear optimization problem in a sparsity-adaptive manner, we use
online mirror descent with a squared $\ell_p$-norm regularizer (with $p$ close to 1), which is well-known to promote
attribute-efficiency in prior works on online learning.

We will add more discussions and explanations on the above points in the final version.

As Reviewer 3 suggests, we will add a brief introduction to online mirror descent in the final version. The extra hard
thresholding step suggested by the reviewer is not necessary (although it would not hurt either - the final error guarantee
would only change by a constant factor); we refer the reviewer to the proof of Theorem 4 for more details.

**References**

[DKTZ20] Learning Halfspaces with Massart Noise Under Structured Distributions, Diakonikolas, Kontonis, Tzamos,
Zarifis, arXiv:2002.05632


[Meta-Review · NeurIPS 2020]

All reviewers agree that this paper made a solid contribution in the context of active learning of sparse halfspaces (in the Massart noise model). The sample complexity bound amounts to a major improvement over best known results for learning of halfspaces, which warrant acceptance of the paper. For camera-ready, the authors are encouraged to take into account the reviewer's feedback to further improve the discussion of the proposed algorithm (In particular, please address the concern on lack of intuition why a mirror descent approach leads to such large improvements in this space compared to earlier methods).